# Review of Data Processing Methods Used in Predictive Maintenance for Next Generation Heavy Machinery

**Ietezaz Ul Hassan, Krishna Panduru * and Joseph Walsh ***

IMaR Research Centre, Munster Technological University, V92 CX88 Tralee, Ireland;
letezaz.ul.hassan@research.ittralee.ie
* Correspondence: krishna.panduru@mtu.ie (K.P.); joseph.walsh@mtu.ie (J.W.)

**Abstract:** Vibration-based condition monitoring plays an important role in maintaining reliable and effective heavy machinery in various sectors. Heavy machinery involves major investments and is frequently subjected to extreme operating conditions. Therefore, prompt fault identification and preventive maintenance are important for reducing costly breakdowns and maintaining operational safety. In this review, we look at different methods of vibration data processing in the context of vibration-based condition monitoring for heavy machinery. We divided primary approaches related to vibration data processing into three categories–signal processing methods, preprocessing-based techniques and artificial intelligence-based methods. We highlight the importance of these methods in improving the reliability and effectiveness of heavy machinery condition monitoring systems, highlighting the importance of precise and automated fault detection systems. To improve machinery performance and operational efficiency, this review aims to provide information on current developments and future directions in vibration-based condition monitoring by addressing issues like imbalanced data and integrating cutting-edge techniques like anomaly detection algorithms.

**Keywords:** predictive maintenance; vibration-based condition monitoring; heavy machinery predictive maintenance; operational safety; machinery performance

## 1. Introduction

Preventive maintenance has the advantage of addressing issues before they become severe, preventing equipment breakdowns [1] and increasing its useful life. This preventative approach provides planned maintenance [2], reducing the rate of unexpected downtime [3] and disturbance for operations. It may still be expensive due to regular maintenance costs [4] and the possibility of over-maintenance, in which elements are replaced early, or maintenance operations are performed unnecessarily. Corrective maintenance [5] saves money from unnecessary maintenance expenses because it is only used when a fault occurs. Unplanned downtime [2] is a drawback of this type of maintenance, and it may have a major negative impact on production and productivity. Meanwhile, if maintenance is set off until a breakdown occurs, severe damage may require more expensive replacements or repairs. Predictive maintenance reduces unplanned downtime, maximises resource allocation [6], and predicts possible issues before they occur using data analytics and monitoring techniques. It provides the advantage of longer equipment life and smaller breakdown. Predictive maintenance systems require advanced technology, and the accuracy of the collected and processed data is important in deciding how reliable the predictions are.

Heavy machinery is specialised equipment used in many different kinds of industries, such as construction, mining, agriculture, manufacturing, or transportation. These machines differentiate themselves by their size, strength, and capacity for performing tasks that are difficult or impossible for smaller machines or humans to accomplish. Drilling rigs, wind turbines, bulldozers, aeroplanes, cranes, excavators, gearboxes, and industry manufacturing machinery are a few examples. For such machinery, predictive maintenance

is needed because of the high cost of downtime, complicated and costly repairs, and safety issues. Predictive maintenance uses sensors, data analytics, and machine learning to monitor performance indicators [7,8], which allows maintenance staff to predict possible defects before they create failures. This proactive method of monitoring helps in the prevention of unplanned downtime, lowers maintenance costs, ensures operational safety, and extends the life of these valuable assets, improving their reliability and performance.

Condition monitoring for heavy machinery includes a number of important parameters [9,10] for monitoring, such as vibration [1,7,11–16] temperature [17], pressure [14,18], or oil analysis. Installing specific sensors [19] and monitoring devices on the machinery, allows for the collection of real-time data [20]. Accelerometers [21] are sensors that are placed properly at certain points on the machine to measure vibrations [22] in different directions and magnitudes. Temperature [17] can be detected in heavy machinery for condition monitoring using specific sensors such as thermocouples or Resistance Temperature Detectors (RTDs) [23]. These sensors transform fluctuations in temperature into electrical signals that can be monitored and analysed. Sensors known as pressure transducers, are used within the device in order to monitor fluid or gas pressure changes. These sensors also convert changes in pressure into electrical signals. These sensors give important data for condition monitoring by converting variations in pressure into electrical signals. Deviations from expected pressure levels [6] may indicate possible problems such as leaks, blockages, or anomalous system behaviour, enabling preventive maintenance and preventing equipment failures. Oil analysis [24] kits or sensors in heavy machinery can be used to measure and collect oil quality data for condition monitoring. These kits or sensors, analyse multiple elements of oil, including metal particle content, deterioration, pollutants, and changes in viscosity. The changes in the oil's quality, including higher levels of pollution or deterioration, could be signs of possible problems related to the machinery.

A number of methods [25] are used to process the data collected for condition monitoring from heavy machinery, including threshold analysis [26], trend analysis [18], statistical analysis [27], frequency analysis, machine learning and predictive analytics. Threshold analysis [21] involves setting a limit on each parameter for detecting deviations from normal values. Trend analysis considers changes in data with time to determine patterns or defects that may indicate possible problems. Statistical technique [28] focuses on data distributions and parameter correlations to find abnormalities or outliers. Frequency analysis [8], which is very helpful for vibration data, allows for identifying specific frequencies associated with mechanical failures. Based on historical data patterns, advanced methods such as machine learning and predictive analytics determine the possibility of failures. Real-time data processing, visualisation, and interpretation can be simplified by integrating failure prediction methods with specialised monitoring systems. These data processing methods make data processing, visualisation, and interpretation straightforward when combined with monitoring systems. Using these methods together allows for more accurate analysis that allows maintenance teams to detect and prevent machine failures [29], optimise operational efficiency and schedule proactive maintenance [20].

Vibration data usually go to a central processing unit, data storage system, or software program after they have been collected. Technicians can then use techniques [30,31] like Fast Fourier Transform (FFT) analysis to break down the vibration signal into frequency components in order to discover specific vibration patterns linked with potential mechanical faults or anomalies. Many [18] procedures and methods are used to process vibration data for condition monitoring in heavy machinery. Time domain analysis [32] analyses the amplitude-frequency [19] and duration of vibration signals using techniques such as peak-to-peak amplitudes, RMS values and waveform analysis. Vibration signals are converted into the frequency domain for frequency domain analysis using the Fast Fourier Transform (FFT), which identifies frequency components associated with weaknesses. Spectral analysis [33] breaks down a signal into its individual frequencies using methods like power spectral density (PSD) analysis. Enveloping [34] extracts high-frequency components for fault detection from vibration data. For in-depth examination,

wavelet analysis [35] divides the signal into many frequency bands. Machine learning and deep learning algorithms provide efficient methods for fault identification in vibration signals. In machine learning, features are usually extracted [13,28,31,36] from the signals and then machine learning classifiers like Random Forests [8,37–39], Support Vector Machines (SVMs) [1,31,40,41] are applied. These classifiers are trained using extracted features, to differentiate between healthy and faulty signals. Deep learning methods, on the other hand, use neural networks [8,17,22,42] with multiple layers to create hierarchical representations from raw data without the requirements of feature extraction. Deep learning models such as Convolutional Neural Networks (CNNs) [1,12,28,41] and Recurrent Neural Networks (RNNs) [41,43] are capable of processing raw vibration signals directly, learning complex patterns and relationships in the data to accurately identify defects.

With the goal of gaining knowledge in the area of condition monitoring for heavy machinery, we have reviewed papers in the literature from 2019 to 2023. The rest of the paper is organised in the given order. Section 2 reviews vibration data processing techniques for condition monitoring. Section 3 discusses the advantages and challenges associated with the methods mentioned in Section 2. Section 4 summarises all processing methods with fine-graining. The conclusion and future directions are given in Section 5.

## 2. Review of Methods

During this study, we looked at 114 research papers, each focusing on the important field of machine condition monitoring, with a special focus on extracting useful information from vibration signals. These articles explained a wide range of signal processing and analysis approaches. They classified the techniques they used as either signal processing, statistics, or artificial intelligence-based. In addition, a number of authors used signal-filtering methods to reduce noise in the signal. Noise reduction is a key step in improving signal quality and helps detect defects in machine signals. The approaches based on artificial intelligence ranged from fuzzy logic-based methods to more advanced feature extraction, selection, and classification methods. In the literature, some articles used statistical parameters to decide whether a signal is faulty or healthy for condition monitoring. Many used machine learning classifiers such as decision trees, random forests, and support vector machines, while others used deep learning techniques such as convolutional neural networks (CNNs) and long short-term memory networks (LSTMs). One-dimensional time series data were immediately fed into 1D Convolutional Neural Networks (CNNs) in a number of papers. On the other hand, some papers chose to use 2D CNNs after converting the one-dimensional time series data into a two-dimensional representation, namely a spectrogram. This methodology variation demonstrates CNNs' flexibility and adaptability in the field of signal processing and analysis. The various methods discussed above show's how many methods are used to achieve efficient heavy machinery condition monitoring. The rest of this section will go deeper into the techniques discussed above.

### 2.1. Signal Processing Based Methods

In particular, some researchers chose to work with these signals in the frequency domain, while others preferred the time domain for their analysis. The remaining group of researchers worked with the time-frequency domain.

### 2.1.1. Time Domain Based Methods

A time domain signal shows variations in a physical quantity (such as voltage, current, vibration, or pressure) over a period of time. In short, it specifies how the signal changes over time. Time domain signals are usually represented as waveforms or graphs, with the x-axis representing time and the y-axis showing the magnitude or amplitude of the signal. Some research papers in the literature focus on the analysis of signals in the time domain. This involves paying attention to the signal's fluctuations or changes over time. This representation indicates how a signal changes with time, making it possible for an understandable knowledge of its behaviour, frequency, amplitude, and duration. The Table 1

below provides a review of time-domain-based approaches, along with corresponding references and explanations of each method's concept.

**Table 1.** Explanation of Fault Identification Methods in Time Domain Analysis.

| Author | Refs. | Method | Description |
|---|---|---|---|
| Wu et al. | [44] | Condition Curve | A condition curve describes the behaviour of a signal over time, and helps in analysing changes in various conditions. |
| Bhuiyan et al. | [45] | Vibration response | Vibration response is known as oscillation or the dynamic movement of a system through time in response to an external excitation or stimulus. |
| Cornel et al. | [46] | Rotation speed range | Rotation speed range is referred to as the range of angular speeds or rotational velocities produced by a system over a certain time period. |
| Morgan et al. | [47] | Vibration analysis | It involves looking into and analysing the behaviour of a system's vibrations over time, which can be utilised to identify defects, monitor performance, or demonstrate dynamic behaviour. |
| Shi et al. | [48] | Operating deflection shapes | Operating Deflection Shapes (ODS) show a system's structural responses during operation. |
| Raghavendra et al. | [49] | Vibration signatures | Vibration signatures are distinct patterns or features of vibrations produced by a system over time which helps in defect identification, condition monitoring, and structural analysis. |
| Marticorena et al. | [50] | Zero Crossing | It describes the points in a waveform where the signal switches polarity and crosses the zero amplitude axis. |
| | | Peak detection | It includes picking out the maximum and minimum points in a waveform for purposes such as event detection. |

Wu et al. [44] used the condition curve of a vibration signal. The authors measured the amount of winding looseness using the Mean Absolute Precision Error (MAPE) and discovered that a gradual increase in MAPE values indicated an increase in winding looseness. Bhuiyan et al. [45] identified faults in signals obtained from a winding machine by analysing the vibration responses. Vibration responses help to detect early warning signs of a malfunction and accurately depict the state of the machinery. Cornel et al. [46] studied the sensitivity of the acoustic emission devices for condition monitoring and determined a rotation speed range based on the size of the AE (Acoustic Emission) amplitude. Rotation speed range improves fault diagnostics under various operating settings by helping to correlate vibration patterns with different operating speeds. A signal or vibration strength or intensity is usually determined by its amplitude, which is the maximum magnitude or height of a wave or oscillation. Morgan et al. [47] used the vibration analysis as a way of monitoring the state of a toggle (A mechanical linkage or system that holds two parts together, usually in a clamped or locked position, is called a toggle clamping mechanism). Vibration analysis helps with preventive maintenance to avert breakdowns by identifying variations in the vibrations of machinery. Shi et al. [48] used Operating Deflection Shapes (ODS) for power transformer winding fault identification. ODS is a technique for visualising and analysing the vibrations of a machine or objects during operation to identify defects and improve performance. They introduced the Histogram of Oriented Gradient (HOG) and Pearson correlation coefficient as approaches by examining the differences between several ODSs. Vibrations of objects during operation can be visualised using Operating Deflection Shapes (ODS). The amplitude of ODS was considered during their decision-making process. Raghavendra et al. [49] predicted defects based on the vibration signature of the signals. A vibration signature refers to the specific pattern of vibrations produced by a particular machine or system, which is often used for condition monitoring and fault diagnostics. These data processing methods were carried out only in the time domain. To keep track of the bearing's structural health, Marticorena et al. [50] examined the kinematics of the bearing cage. They used a digital low-pass filter after using the angular

resampling technique. They then performed zero-crossing detection and peak detection. By detecting variations in vibration frequency and timing that could indicate faults in the machinery, zero crossing helped in identification of faults. Peak detection can be helpful for detecting anomalous vibration amplitudes, which allows for the early diagnosis of issues such as imbalance. They were able to estimate the shaft rotation between detections of the same pocket due to these procedures, which they used for condition monitoring of bearings.

2.1.2. Frequency Domain Based Methods

A vibration signal is represented in the frequency domain by the frequencies that make it up. A signal's frequency domain representation indicates energy distribution across different frequencies. Therefore, we study the different individual frequencies in the signal and their amplitudes rather than how a signal changes over time. It is a method of understanding the many building blocks of a complex signal, which may be highly helpful for various data analysis, compression, and filtering tasks. Skowronek et al. [51] focused on identifying background noise and found that it primarily exists in high-frequency components. Therefore, they attempted to reduce this noise because removing it can help select appropriate tools for signal analysis. Nirwan et al. [52] used vibration analysis to forecast rolling mill faults by processing signals. They predicted defects by determining the highest frequency in the frequency spectrums. A signal's high-frequency component can identify fine-scale fluctuations such as gear meshing problems or bearing failures. Pavan Kumar et al. [53] used the peak value obtained from the signal's frequency spectrum as a decision criterion for fault diagnoses. The peak value computed from a signal's frequency spectrum shows anomalous amplitudes at certain frequencies, which could be signs of a fault such as gear damage or bearing issues. Mehamud et al. [54] observed defects in the signals obtained from a TENG (Triboelectric Nanogenerator) by examining the signal's frequency spectrum. A signal's frequency spectrum can be used to examine changes in spectral properties, such as peaks or shifts, which may indicate fault signs associated with misalignment, imbalance, or bearing wear in equipment. Gurusamy et al. [55] focused on the magnetic flux spectrum of signals for fault identification. They pointed out that the magnetic flux spectrum gives more information about faults compared to current signatures. The magnetic flux spectrum directly reflects the mechanical state of rotating machinery, thus, it allows for insight into phenomena that may not be as easily seen via electrical signals alone, such as rotor eccentricity, variations in the air gap, and structural deformations. Spectrum is the distribution pattern of amplitudes or energy of different frequencies within a signal. Wang et al. [56] developed a technique for monitoring a cat-head tower's condition based on vibration responses. They measured the differences in the frequency response function's X, Y, and Z directions. The gaps in the frequency response function correspond to the system's resonant frequencies or modes, since they may increase specific vibrations, which can lead to physical damage or failure. They noted the result as invalid if the maximum difference reached above a particular threshold. Hu et al. [57] developed a novel method for concurrently capturing and analysing vibration signals, with the goal of enhancing signal quality by measuring instantaneous shaft speed from the spectrogram (a graphical representation of the frequency spectrum of a signal over time) and using it for data resampling. Tracking changes in shaft speed over time allows one to detect variations in machine performance, such as speed fluctuations, load changes, or anomalies caused by issues like imbalance or misalignment. Priebe et al. [23] started pre-processing their data using various signal-processing techniques. Slicing the data, removing the moving average from the signal, applying the Hilbert transform, and transferring the signal to the frequency domain were the approaches they used. They used damage indicators to determine the faults. Peters et al. [58] examined the amplitude spectra to find defects in the vibrational signals. Amplitude spectra provide information about the intensity or power of each frequency inside a signal by displaying the amplitudes or magnitudes of the various frequency components. Peeters et al. [59] developed a unique blind filtering method

that used the envelope spectrum's sparsity to identify defects in vibration signals with a second-order cyclostationary signature. Envelope spectrum sparsity of vibration data can be used to identify defects by looking for abnormalities in spectral peak distributions. The sparsity of the envelope spectrum relates to how many of its components are non-zero. Table 2 revises the techniques for condition monitoring based on signal processing in the frequency domain.

**Table 2.** Explanation of Fault Identification Methods in Frequency Domain Analysis.

| Author | Refs. | Method | Description |
|---|---|---|---|
| Priebe et al. | [23] | Damage signatures | A damage signatures in the frequency domain are distinct patterns that correspond to anomalies or damage. |
| Skowronek et al. | [51] | High Frequency Component | The higher frequency portions of the signal. |
| Nirwan et al. | [52] | High Frequency Component | The higher frequency portions of the signal. |
| Pavan Kumar et al. | [53] | Peak Value from Frequency Spectrum | It shows the highest possible amplitude or magnitude of a particular frequency component. |
| Aburakhia et al. Rajapaksha et al. Mehamud et al. | [5] [39] [54] | Analysis of Frequency Spectrum | It involves analysing the properties and distribution of the frequency components that make up a signal. |
| Gurusamy et al. | [55] | Magnetic Flux Spectrum | It provides details about the magnetic characteristics and behaviour of the signal by representing the distribution and characteristics of magnetic flux density across various frequencies. |
| Wang et al. | [56] | Differences in frequency response function | It indicates differences in the frequency responses of two signals or systems, which indicate differences in the behaviour of the systems. A frequency response function is a measurement of how a system reacts to an input vibration at various frequencies. |
| Hu et al. | [57] | Instantaneous shaft speed from Spectrogram | It is the rotational speed that varies over time as a result of changes in frequency content. |
| Peters et al. | [58] | Amplitude spectra | It shows the relationship between the amplitude of the various frequency components that make up the signal. |
| Peeters et al. | [59] | Envelope spectrum sparsity | In the context of the total number of frequency bins, it shows how tightly packed the important peaks are in the envelope. |

### 2.1.3. Time and Frequency Domain Based Methods

In signal analysis, processing signals within the time and frequency domains [42,51] is a popular and effective method. It gives more information for understanding the signal's properties. The frequency domain shows the signal's fundamental frequency components, whereas the time domain illustrates how the signal changes over time. Sometimes the dual-domain method works well to acquire specific knowledge and characteristics required by different applications. The two domains can be easily converted using methods like Fourier Transforms. A combination of time-frequency representation is provided by specialised techniques such as the Continuous Wavelet Transform or Short-Time Fourier Transform, which offer important insights into the temporal evolution of frequency components. Mystkowski et al. [60] focused on using affordable accelerometers for condition monitoring in their research. They compared their proposed model for condition monitoring using amplitude and phase signals. They found a linear relationship between RMS acceleration values and rotational speed, indicating that one-second subsets are enough for signal processing. RMS to speed ratios vary among machine points, showing the importance of frequency approaches to enhanced machine component diagnostics. Zhao et al. [61] used frequency and amplitude of vibration data from a VS TENG (Vibration-based Triboelectric

Nanogenerator) to monitor the health of machinery. Civera et al. [16] proposed the use of Instantaneous Spectral Entropy and Continuous Wavelet Transform for anomaly detection and fault diagnostics using vibration data from gearboxes in their research. Instantaneous spectral entropy improves the identification of vibration patterns during vibration-based condition monitoring by evaluating the complexity or unpredictability of the frequency spectrum over time. Continuous wavelet transform (CWT) improves the detection of vibration patterns by providing a time-frequency representation of the signal, allowing the identification of transient events, frequency modulations, and time-localised features related to defects. Morgan et al. [21] presented an unsupervised model for signal denoising in their paper. Their model used a hybrid of continuous wavelet transform and singular value decomposition techniques. Singular value decomposition (SVD) allows for the detection of vibration patterns by splitting down a signal into its patterns or features, which allows for the detection of patterns that correspond to a defect. They also used the Inverse Continuous Wavelet Transform in their model. Through spectral analysis, Michalak et al. [33] could determine the highest-energy deterministic component, estimate its frequency and amplitude from the spectrum, and subtract it from the input signal. They determined the faulty signals based on the highest energy components. Table 3 revises the methods in the time-frequency domain used for vibration-based condition monitoring, which is reviewed above.

**Table 3.** Explanation of Fault Identification Methods in Time-Frequency Domain Analysis.

| Author | Refs. | Method | Description |
|---|---|---|---|
| Civera et al. | [16] | Instantaneous spectral entropy, Continous wavelet transform | The degree of uncertainty of a signal frequency spectrum at a given point in time is measured by instantaneous spectral entropy. The continuous wavelet transform breaks down a signal into wavelet coefficients, which are then used for temporal frequency analysis. |
| Morgan et al. | [21] | Hybrid of CWT and SVD | By breaking down a signal matrix into its three constituent matrices, singular value decomposition provides information about the dominant mode, variability, and signal structure. |
| Michalak et al. | [33] | Highest energy component | The frequency or component of a signal with the highest amplitude or energy contents is referred to as having the highest energy component. |
| Mystkowski et al. | [60] | Amplitude and Phase | The maximum distance from the centre line to the peak is known as the amplitude. Any particle in a waveform that shows periodic behaviour can be identified by its phase. |
| Zhao et al. | [61] | Frequency and Amplitude | The number of times something occurs in a particular period of time called its frequency. |

### 2.1.4. Preprocessing Methods for Improving Signals Quality for Fault Detection

The ability to extract relevant information from collected vibration signals while reducing the negative impacts of noise is an important aspect of diagnosing faults accurately. Signal processing filters are important to improve the quality of vibration data and the accuracy of machine fault identification. To reduce the noise in the recorded vibration signals, Ghazali et al. [62] used a Kalman filter. Kalman filters are applied to optimally measure variables of interest when they are impossible to measure directly but have an indirect measurement available. It is commonly implemented in signal processing and control systems in order to reduce noise while improving the accuracy of a signal. Sharma et al. [63] used the Fast Fourier Transform to look at the signals in the frequency domain, and spectrograms to evaluate the signals in the time-frequency domain. They used a chain of low-pass filters combined with the Hilbert Transform (HT) to improve frequency resolution and remove noise from these vibration signals. They also refined the vibration signals with several filters, such as low-pass sparse banded filters, Savitzky–Golay

differentiator filters, and variational filters. Krot et al. [64] used passband filters to select natural frequencies and higher harmonics for their study, analysing changes from previous observations for condition monitoring. Passband filters work well for separating specific frequency bands associated with equipment failures, which allows a more detailed study and identification of fault characteristics while reducing noise and interference of other frequencies. The AR+Noise technique (a variant of the standard AR parametric method) was created to address noise levels in real data by Zonzini et al. [65]. Auto-Regressive plus Noise is a signal processing method that uses an autoregressive model with an additional noise component for modelling and analysing time-series data. AR+Noise modelling breaks down deterministic components from random fluctuations, which allows for more accurate fault diagnosis in noisy data. Rafiq et al. [36] proposed NAMEMD, an improved MEMD method, which involves adding white Gaussian noise that is uncorrelated to multivariate signals to assist in extracting useful multivariate IMFs. This method successfully handles multi-channel vibration signals that have high speed/load variations while reducing mode mixing. Customised signal processing methods are frequently developed by some authors for more specialised and accurate defect diagnosis in a range of applications. Bhowmik et al. [2] developed a condition monitoring model relying on a single-sensor-based output-only algorithm in their work. This method was built using the first-order Eigen perturbation (FOEP) technique and Recursive Singular Spectrum Analysis (RSSA). FOEP gives information about the sensitivity of a system's dynamic performance to slight changes in its parameters, making it easier to identify important components and possible fault modes in equipment. Recursive Singular Spectrum Analysis (RSSA) allows the real-time breakdown of vibration signals for trend analysis and fault diagnosis. At every point in time, their suggested algorithm updated the Eigenspace. Fernando et al. [66] used vibration data collected from bridges for condition monitoring and applied the Eigensystem Realisation Algorithm. The Eigensystem Realisation Algorithm predicts a state-space model based on impulse response data instead of input/output. Patil et al. [67] put up a fusion-based REB (Rolling Element Bearing) model that blends the multi-body dynamic model and the acoustic emission model. Lin et al. [68] compared the outcomes of two identification algorithms—frequency-domain decomposition (FDD) and frequency spatial decomposition (FSDD)—and they carried out a comparative study. Frequency-domain decomposition (FDD) is a technique for breaking down a signal into its frequency components, allowing for the analysis and characterisation of the signal's frequency content as well as spectral properties. Frequency spatial decomposition is a technique that breaks down a signal into its frequency components while also taking spatial fluctuations into consideration. This consideration allows for simultaneous analysis and characterisation of signals in both the frequency and geographical domains. Frequency spatial decomposition and frequency domain decomposition allow for the separation of various frequency components of signals for analysis, which allows for identification of defects signs, spatial variations and mode forms related to faults. Yin et al. [15] used a three-dimensional approach to compress the signal, with the three dimensions referring to length, width, and height. It plays a role in recording and representing the multidimensional features of vibration data, allowing for more accurate analysis, and feature extraction.

### 2.2. Statistics Based Methods

Statistics-based techniques are methods that use statistics concepts and ideas to analyse and interpret data. Kamariotis et al. [69] performed structural health monitoring of synthetic data using the Bayesian decision analysis model. Nithin et al. [10] evaluated the peak component, crest factor, power, RMS value, and acceleration character spectrum of vibration signals for bearing defect detection and compared them to signals from good bearings. Rahman et al. [70] used a vibration analysis to determine fault using the signal's amplitude, phase angle, and frequency. Han et al. [71] used remaining useful life for predicting the health of the manufacturing system. The technique provides a mission reliability-oriented RUL prediction method and redefines failure modes by taking into ac-

count the functional dependence of components on product quality. Li et al. [72] presented an integrated predictive maintenance (PdM) method for manufacturing systems, with a focus on the negative effects of product faults. They introduced and applied two kinds of dependability metrics to PdM tasks. The best PdM approach minimised overall costs and was selected after assessing maintenance costs and financial loss.

### 2.3. Artificial Intelligence Based Methods

The arrival of artificial intelligence (AI) has brought about an entirely new age of structural health monitoring and industrial machinery maintenance. AI-powered techniques have proven useful in processing vibration signals with remarkable precision and efficiency, opening up an entirely novel field for fault detection. Machine learning and deep learning are artificial intelligence subfields.

### 2.3.1. Machine Learning Based Methods

Machine learning is a type of artificial intelligence that deals with developing algorithms and models that allow computers to acquire patterns and make predictions based on patterns identified in data without being manually programmed. Machine learning models can be supervised (used for classification) or unsupervised (used for clustering). Machine learning cannot work with raw data; to train machine learning models, important features must first be extracted from raw vibration signals, which are then used to train the models. Different types of useful features can be extracted from the raw vibration data, such as statistical features. Feature selection approaches are used to choose the most significant features and thereby reduce dimensionality. Table 4 below provides a summary of the literature in our study, with a focus on research that uses machine learning methods. Table 5 contains an explanation of machine learning classifiers.

**Table 4.** An overview of feature extraction, selection, and feature classification methods in vibration-based condition monitoring.

| Author | Refs. | Feature Extraction | Feature Selection | Feature Classification |
|---|---|---|---|---|
| Kannan et al. | [9] | - | Majority Voting | One Class SVM |
| Wang et al. | [13] | FAST<br>SIFT | - | Codebook vectors<br>Word frequency vectors |
| Nowakowski et al. | [30] | - | - | Decision tree |
| Li et al. | [31] | OSE | - | SVM |
| Mazzoleni et al. | [34] | HMM<br>OCSVM<br>Envelope analysis | - | Fuzzy model |
| Sharma et al. | [37] | - | -<br>KNN | Random forest |
| Balachandar et al. | [38] | Statistical features | Decision tree | Random forest |
| Brusa et al. | [73] | - | SHAP | SVM, KNN |
| Sun et al. | [74] | Statistical features +<br>Variational model<br>decomposition energy<br>entropy features | ReliefF | SVM |
| Joshuva et al. | [75] | Histogram features | J48 decision tree | Locally weighted<br>learning model |
| Patange et al. | [76] | Statistical features | - | Random forest |

**Table 4.** *Cont.*

| Author | Refs. | Feature Extraction | Feature Selection | Feature Classification |
|---|---|---|---|---|
| Pranesh et al. | [77] | Statistical features | - | Logistic algorithm |
| Patange et al. | [78] | Statistical features | - | Decision tree<br>Ensemble method |
| Harish et al. | [79] | Statistical features | - | Logit boost |
| Balachandar et al. | [80] | Statistical features | - | Best first tree |
| Lipinski et al. | [81] | - | - | Decision tree |
| SUN et al. | [82] | - | Fisher discrimination<br>Relief | SVM |
| Mukherjee et al. | [83] | - | - | Ensemble Model |
| Gómez et al. | [84] | 6 wavelet variance features<br>16 shannon entropy features | - | SVM |
| Tsunashima et al. | [85] | - | - | SVM |
| Lu et al. | [86] | -<br><br>Extreme boosting | -<br>SVM | Neural network |

**Table 5.** Specifications for Machine Learning Classifiers.

| Classifier | Description |
|---|---|
| Logistic Regression | Logistic Regression is a binary classification technique that predicts the probability of a binary classification problem using one or more independent variables and a logistic function to estimate the likelihood of the occurring outcome. |
| Decision Tree | The decision tree algorithm splits the feature space into a tree-like structure and makes decisions based on a number of simple binary splits, giving easy visualisation and comprehension of how decisions are made. |
| Random Forest | Random Forest is a form of ensemble learning that creates several decision trees and combines the results they produce in order to improve robustness and accuracy. It prevents overfitting, which makes it useful for regression and classification problems. |
| Naive Bayes | Naive Bayes refers to a statistical classification algorithm that relies on Bayes' theorem and assumes that features are conditionally independent. It is especially useful for high-dimensional data. |
| SVM | The support vector machine (SVM) is a supervised learning method that is used for regression and classification tasks. It determines the best hyperplane that separates various classes by maximising the margin between them, making it suitable for both linear and non-linear classification problems that use kernel functions. |
| One Class SVM | The One-Class Support Vector Machine (OCSVM) is an improved form of the traditional SVM method that can be used for outlier detection tasks with only one class of data available for training. It creates a hyperplane that captures most data points while reducing outliers, making it suitable for identifying deviations in unlabeled datasets. |
| KNN | The K-Nearest Neighbours (KNN) algorithm is a method for classification that predicts the majority of the class of its k nearest neighbours in feature space. The parameter k specifies the number of neighbours to take into account, KNN assigns the class label of the majority to the k neighbours to the new instance. Although KNN is easy to use, but it is computationally expensive especially for large datasets because it requires computing power to calculate the distances between each new instance and each training instance. In contradiction to this, KNN is frequently used for classification tasks, especially in situations where interpretability is of great importance or when the dataset is small. |
| Ensemble Model | An ensemble model is an approach in machine learning that aggregates predictions from many independent models to improve overall performance. It combines the predictions of various base models, such as decision trees, support vector machine, and neural network, to give an overall prediction that usually produces much better results than any individual model. |

Gildish et al. [19] have compared the performance of three machine learning regressors in their paper: Ridge, support vector regressor, and deep learning regression. The support vector regressor yielded excellent results. Kannan et al. [9] acquired information from Spectra Quest machinery did condition monitoring on individual sensors, weighted their output, and then aggregated it for the final decision using the majority vote approach. Majority voting leverages individual, collective expertise to increase overall accuracy and precision. A one-class support vector machine was used to produce the final classification. Because one-class support vector machines perform well at detecting abnormalities or deviations in normal patterns within unlabeled data. Brusa et al. [73] used Shapley Additive Explanation (SHAP) as a feature selection method on data gathered from medium-sized industrial bearings. SHAP makes it possible to select features by ranking their importance with respect to model predictions, making it easier to determine the most useful features for condition monitoring. They used SVM and KNN to classify the signals for monitoring conditions. Sun et al. [74] used a combination of approaches for fault diagnosis in their study, combining both signal processing techniques and artificial intelligence-based methods. They combined the time and frequency domain statistical features with variational model decomposition energy entropy features. Statistical features are useful because they provide useful information about the statistical characteristics of vibration signals, which allows for the analysis of vibration patterns, the identification of anomalies, and the detection of faults using measures such as variance, kurtosis and mean. Intrinsic Mode Functions (IMFs) are the basic elements of the Empirical Mode Decomposition (EMD) approach, with periodic modes having clearly defined frequency properties and zero mean envelopes. They used ReliefF as a feature selection approach and Support Vector Machine (SVM) to classify these features into faults. ReliefF evaluates the relevance of features by looking at the weights assigned to them according to their ability to differentiate between instances of different classes. Joshuva et al. [75] obtained vibration signals from wind turbines and used these signals to extract histogram features. Histogram features are features used to describe the distribution of data values into intervals or discrete bins. They used a locally weighted learning model for classification and the J48 decision tree for feature selection. The J48 decision tree algorithm is also used as a method for choosing features by looking at the attribute importance scores generated by the tree structure. Patange et al. [76] collected vibration data from a standard lathe machine, extracted statistical features from these signals and used the random forest to classify these signals. Pranesh et al. [77] extracted statistical features from the brake vibration signals and classified them using a logistic algorithm. Patange et al. [78] extracted statistical features from the acquired Industrial VMC machine data and classified them using a decision tree. They also investigated combining decision trees into an ensemble method. Harish et al. [79] collected vibration signals from an automobile LMV's hydraulic brake, extracted statistical features from them, and subsequently utilised the Logit Boost algorithm for classification. Balachandar et al. [80] collected vibration signals from the FSW tool conditions, extracted statistical features from these signals, and then used the Best First Tree classifier to classify them. Lipinski et al. [81] used a spectral representation of the time-domain segmented raw signal to represent the features extracted from the gearbox as a vector with a size of 15. The spectral representation of a signal provides extracted features by recording the frequency information of the signals within certain time segments. They used a decision tree to classify the data. SUN et al. [82] proposed a more precise feature extraction method by combining multiscale fluctuation-based dispersion entropy and variable mode decomposition. The multiscale fluctuation-based dispersion entropy (MFDE) computes the dispersion entropy of a variation series acquired by dividing the original signal into different scales with a wavelet transform or similar multiscale decomposition techniques. A two-stage feature selection strategy based on Fisher discrimination and Relief was also proposed. An SVM classifier was used to diagnose faults caused by vibration signals. Sharma et al. [37] evaluated the performance of several types of features using Random Forest and KNN. Mukherjee et al. [83] used an ensemble model to classify the vibration signals obtained from the blender after removing noise by using basic

time and frequency domain operations. Balachandar et al. [38] extracted statistical features from the collected vibration data. The performance of four different machine learning classifiers was evaluated, with the decision tree as the feature selection method. Among the J48 (decision tree), Hoeffding, LMT and Random Forest classifiers, the Random forest showed the best results. Nowakowski et al. [30] first decomposed the gathered signals by using the EMD method and then classified them using the decision tree for the identification of faults. Li et al. [31] perform classification using SVM after first extracting features from the raw signals using OSE (optimal symbolic entropy). Optimal Symbolic Entropy (OSE) method measures the complexity of a signal by transforming it into symbolic sequences and calculating their entropy. OSE determines the appropriate signal split into symbols that maximise entropy. Gómez et al. [84] evaluated the performance of several classifiers by extracting features from the time domain, frequency domain, and time-frequency domain data. Six wavelet variance estimation features and sixteen Shannon entropy features were considered. Wavelet variance estimation features are statistical characteristics based on the variance of wavelet coefficients obtained via wavelet transform analysis. These features represent the variability and changes in signal properties at various scales, giving useful information for identifying issues and trend analysis. Shannon entropy features are statistical parameters derived from Shannon entropy that measure the degree of unpredictability or predictability of a random variable's distribution. In particular, the support vector machine outperformed the other classifiers. Tsunashima et al. [85] used SVM for fault detection from synthetic data. Lu et al. [86] presented a data-driven method to recognize early warning signs in mill vibration signals. They evaluated how well three different approaches—neural network-based, support vector machine-based, and extreme gradient boosting—performed at detecting faults in the mill's recorded vibration data. Three different feature extraction techniques- entropy features, linear discriminant analysis, and artificial neural network, were combined by Wang et al. [13]. Combining these features can improve the quality of feature representation by gathering additional information with respect to discriminative power, nonlinear relationships and signal complexity. They first transformed the collected signals into grayscale images and then extracted the features using the FAST and SIFT methods. They used codebook vectors and word frequency vectors to detect faults.

Unsupervised machine learning algorithms are used when labelled data are not available. The task is to cluster the data points so that they are very similar within a group and dissimilar from the data points of another group. Hendrickx et al. [32] start with the machine performance data in their paper and compare, and cluster them appropriately. They use an unsupervised method to assign anomaly scores to identify machine faults. Nie et al. [87], for monitoring the structural health of bridges, developed a fixed moving principal component analysis method for analysing vibration data from the bridges. Mazzoleni et al. [34] collected temperature, pressure, current, and voltage data from the process. After extracting features from these time series data, they examined the health indicator using several methods such as HMM, One-Class SVM, and Envelope analysis. They used the Fuzzy model for the identification of faults.

### 2.3.2. Deep Learning Based Method

Machine learning has limitations, like challenging feature engineering, limited performance, and generalisation difficulties. The following are some issues related to traditional machine learning that have been solved by deep learning.

1.  Feature Engineering: Deep learning simplifies feature extraction, thus decreasing the need for manual feature engineering.
2.  Scalability: Deep learning models are much more efficient at handling huge datasets than typical machine learning methods.
3.  Non-linearity: Deep learning architectures can more effectively capture complicated, nonlinear relationships in data.
4.  Representation Learning: Algorithms using deep learning can acquire hierarchical data representations, that allow them to generalize more effectively to new tasks.

5. End-to-End Learning: Deep learning models may learn directly from raw data, reducing the need for constructed pipelines to extract features in standard machine learning.

Deep learning comprises many different models, each with its own set of properties. Deep learning's base is artificial neural networks, which mimic the brain's information processing units and are inspired by the human brain. Popular deep learning models are mentioned in Table 6 with some explanation. Dabrowski et al. [22] demonstrated the hardware implementation of a neural network for planetary gearbox condition monitoring in their paper. They gathered planetary gearbox vibration signals. Elvira-Ortiz et al. [18] used an artificial neural network for performing the classification after combining the obtained features. They extracted entropy-related features, which were then selected by linear discriminant analysis (LDA). Zonzini et al. [88] used an artificial neural network to detect faults in the data collected from the X24 bridge, followed by signal compression using the MRAK-CS approach. They used a graph convolutional neural network as a feature extractor. Espinoza Sepúlveda et al. [89] created rotor fault simulations and collected data from them. They extracted features such as the root mean square value, variance, skewness, and kurtosis. They proposed a smart vibration-based machine learning model (SVML) based on a Multi-Layer perceptron for classifying these feature vectors in order to find defects. Espinoza-Sepulveda et al. [24] used a Multi-Layer perceptron in their study to detect faults in vibration signals. They also extracted statistical features from signals. Sepulveda et al. [90] evaluated time domain features such as root mean square, variance, skewness, and kurtosis for fault identification from vibration signals. They presented a vibration-based machine learning model for fault identification based on multi-layer perceptron. Inturi et al. [91] collected 13 health-related features from the raw vibration data gathered from the gearbox and classified them using both traditional machine learning and deep learning. Demircan et al. [92] utilised features collected from the time domain, frequency domain, and time-frequency domain signals to evaluate the performance of several classifiers. Six wavelet variance estimation features and 16 Shannon entropy features were extracted. They used artificial neural networks to classify these features. Table 7 reviews deep learning-based methods for vibration-based condition monitoring used in the literature.

**Table 6.** Explanations of Deep learning models.

| Classifier | Description |
| --- | --- |
| ANN | A computational model inspired by the human brain's composition and operation is known as an artificial neural network (ANN). An input layer, one or more hidden layers, and an output layer make up the layers that consist of interconnected nodes or neurons. Several learning algorithms, such as backpropagation, train ANNs to detect complex patterns and relationships in data. ANNs are often used to detect faults and evaluate conditions by learning complex patterns and correlations from vibration data. |
| CNN | A convolutional neural network (CNN) is a form of artificial neural network used to interpret structured grid data like images or signals. It comprises several layers, such as convolutional, pooling, and fully connected layers. CNNs are very useful for tasks such as image classification and object detection because they learn hierarchical features from raw data. After extracting local features from the input data using convolutional filters, they reduce the spatial dimension by pooling layers and improve computational efficiency. CNNs can analyse vibration data in the time-frequency domain and extract hierarchical features to detect faults and abnormalities in machinery. |
| RNN | An artificial neural network type called a recurrent neural network (RNN) is made up of processing sequential data, such as natural language data or time series data. In contrast to feedforward neural networks, RNNs include connections that create directed cycles, allowing them to show dynamic temporal behaviour while maintaining an internal memory of prior inputs. This makes it possible for RNNs to represent sequential dependencies and long-range dependencies in data in an efficient manner. Yet, they have issues like vanishing gradients and difficulties in capturing long-term relationships, which lead to the creation of more advanced architectures such as Long Short-Term Memory (LSTM) and Gated Recurrent Unit (GRU) networks. RNNs can detect temporal dependencies in vibration data, making them useful for time series analysis, predicting future states, and detecting anomalies based on past data. |

**Table 6.** *Cont.*

| Classifier | Description |
|---|---|
| GAN | A neural network architecture known as a Generative Adversarial Network (GAN) consists of two networks, a discriminator and a generator, that are trained together by using a competitive process. The generator is trained to generate synthetic data samples that resemble actual samples from a specified distribution, and the discriminator is trained to distinguish between synthetic and real samples. Adversarial training improves the generator's ability to produce realistic samples, while the discriminator improves its ability to distinguish between actual and fake samples. Generative Adversarial Networks (GANs) can produce synthetic vibration data samples representing real-world situations, helping with data augmentation and producing different datasets for training machine learning models in vibration-based condition monitoring. |

**Table 7.** Overview of Deep Learning Techniques for Vibration-Based Condition Monitoring.

| Author | Refs. | Feature Extraction | Feature Selection | Feature Classification |
|---|---|---|---|---|
| Ong et al. | [1] | - | - | CNN |
| Amin et al. | [4] | - | - | CNN |
| Koutsoupakis et al. | [12] | - | - | 1D-CNN |
| Civera et al. | [16] | - | - | CNN |
| Murgia et al. | [17] | - | - | CNN |
| Elvira-Ortiz et al. | [18] | Entropy features | LDA | ANN |
| Afridi et al. | [20] | - | - | LSTM |
| Dabrowski et al. | [22] | - | - | ANN |
| Espinoza-Sepulveda et al. | [24] | Statistical features | - | MLP |
| Wang et al. | [27] | - | - | ANN |
| Chesnes et al. | [28] | Linear and Quadratic Discriminant Analysis | - | CNN |
| Vos et al. | [40] | - | - | LSTM-OCSVM |
| Koutsoupakis et al. | [42] | - | - | CNN |
| Zonzini et al. | [88] | Graph convolutional neural network | - | ANN |
| Espinoza Sepúlveda et al. | [89] | Statistical features | - | MLP |
| Sepulveda et al. | [90] | Statistical features | - | MLP |
| Inturi et al. | [91] | Hurst exponent | - | DNN |
| Demircan et al. | [92] | Shannon entropy Wavelet variance estimation | - | ANN |
| Yaghoubi Nasrabadi et al. | [93] | Frequency response function | LASSO | CNN |
| Amin et al. | [94] | - | - | CNN |
| Naveen Venkatesh et al. | [95] | - | - | VGG16 Alexnet ResNet50 GoogleNet |
| Ahmad et al. | [96] | - | - | LSTM based Auto encoder |
| Feng et al. | [97] | - | - | GRU |
| Hong et al. | [98] | - | - | GRU |
| Zhao et al. | [99] | - | - | GAN |

**3. Challenges, Advantages in Data Processing Methods for Vibration-Based Condition Monitoring**

*3.1. Relationship between Investments in Heavy Machinery and the Importance of Vibration-Based Condition Monitoring*

Heavy machinery investments usually involve considerable financial investments, and ensuring the best performance of assets for maximising return on investment. Vibration-based condition monitoring is important in this regard because it provides information about the machinery's health and condition. Below is how they relate.

1. Early Detection of Faults: Early indications of mechanical issues, such as imbalance, misalignment, inequalities, bearing wear, and structural flaws in heavy machinery, can be found by vibration analysis. Early detection of these problems allows for proactive scheduling of maintenance [100], which reduces downtime and saves expensive breakdowns.

2. Optimised Maintenance Strategies: Maintenance teams can apply condition-based maintenance plans by using vibration analysis for condition monitoring. Rather than maintaining fixed maintenance schedules [101], which can result in excessive downtime and costs, maintenance may be performed whenever the equipment's condition confirms it is required. This method increases the life of the machinery and decreases maintenance expenses.

3. Improved Safety: Vibration monitoring helps in the identification of possible safety risks [102], such as high vibration levels which indicate that the breakdowns are going to happen. It is possible to reduce the risk of accidents and injuries related to running heavy machinery by taking immediate measures on these issues.

4. Enhanced Performance and Efficiency: Operators are able to adjust machinery operation for maximum performance and energy efficiency through routine vibration level monitoring. Machinery may operate more smoothly, using less energy and operating more efficiently overall, by detecting and repairing anomalies in vibration patterns.

5. Data-driven Decision Making: Vibration data recorded over time give important insights into the health trends and long-term performance of heavy machinery. Stakeholders can make well-informed decisions on equipment replacements, upgrades, and asset management plans by analysing these data [103].

*3.2. Challenges in Signal Based Processing Methods*

Using signal processing techniques for vibration-based condition monitoring raises a number of challenges. Advanced signal processing methods and domain expertise are needed to address these challenges.

1. Noise and Interference: The extraction of relevant information from vibration signals can be difficult since they often are polluted by noise and interference from several sources, such as electrical noise, background vibrations, or mechanical vibrations from nearby machinery [104].

2. Non-Stationarity: In real-world situations, vibration signals often have non-stationary behaviour, which means that with time, their statistical characteristics fluctuate. Because of this non-stationarity [105], vibration data analysis and interpretation are more difficult, which requires the use of signal processing techniques.

3. Complexity of Fault Signatures: Particularly in the early phases of fault development, fault signatures in vibration signals can be unclear and tricky. Advanced signal processing methods that can pick up hidden trends are needed to identify these signatures [106] between operational fluctuations and background noise.

4. Feature Extraction and Selection: It can be difficult to extract relevant characteristics from vibration signals and choose those that are most helpful for defect diagnosis and detection. For designing effective feature extraction methods [104], that are appropriate to certain machinery and defect types, domain knowledge is needed.

5. Data Volume and Dimensionality: Large volumes of high-dimensional data [107] are produced by vibration-based condition monitoring, which creates difficulties in

processing, storing, and analysing the data. To efficiently process huge amounts of data, efficient methods for data compression, real-time processing and dimensionality reduction are required.

### 3.3. Challenges in Artificial Intelligence Based Processing Methods

While using AI-based approaches for vibration-based condition monitoring, many challenges arise during the feature extraction, selection, and classification phases:

1.  Feature Extraction Complexity: Extracting useful features [108] from vibration signals can be complicated. Developing effective feature extraction methods that capture useful information while reducing noise and irrelevant data can be difficult.
2.  Dimensionality Reduction: High-dimensional feature spaces can result in the curse of dimensionality [109], which occurs when the number of features exceeds the number of observations, increasing complexity, overfitting, and reducing generalisation performance. To address these issues, dimensionality reduction techniques must be used.
3.  Feature Selection Bias: The potential bias generated by feature selection algorithms makes it difficult to choose the best discriminative features [110] for defect detection and classification. Robust performance requires that the features chosen represent a variety of fault patterns and operate well under different operating scenarios.
4.  Data Imbalance: Unbalanced datasets could arise due to the few samples of defective conditions [111] in vibration-based condition monitoring as compared to normal operating conditions. Unbalanced datasets may introduce bias into the learning process and lead to poor classification performance, causing the need for the use of specialised techniques such as ensemble methods, resampling, or cost-sensitive learning.
5.  Interpretability: AI models, especially deep learning models, are commonly referred to as black boxes due to their complex topologies and nonlinear transformations. Understanding the conclusions made by these models and interpreting the underlying principles behind defect detection and classification can be difficult, minimising their use in safety-critical applications where interpretability is important [112].
6.  Generalisation to New Fault Types: Artificial intelligence models trained on historical data might have difficulty while generalising to new defect types or operational situations that did not occur in training. The careful evaluation of data representation, model architecture, and training procedures must be performed to ensure AI models' robustness and generalisation capabilities across various fault scenarios.
7.  Computational Resources: Large-scale datasets and high-performance computer systems are two of the most important computational resources [108] needed for training deep learning models. Organisations with limited funding or infrastructure may have issues accessing enough computational resources.

### 3.4. Advantages of Signal Based Processing Methods

Signal processing methods provide advantages in terms of interpretability, computational efficiency, domain knowledge integration, data availability robustness, and modelling pipeline transparency. These advantages make signal-processing methods useful tools for vibration-based condition monitoring, particularly for applications requiring interpretability and domain knowledge.

1.  Interpretability: Signal processing methods often generate interpretable results [113], allowing experts in the field to better understand the underlying dynamics for fault detection and diagnosis. AI models, particularly deep learning models, are frequently referred to as black boxes due to their complicated architectures and nonlinear transformations, which make interpretation difficult.
2.  Computational Efficiency: Signal processing methods require a smaller amount of computer resources than AI-based methods, making them better suited for applications with limited real-time processing and computational infrastructure units [114].



Signal processing techniques can be developed and implemented using lightweight techniques that perform well on edge devices and embedded systems.

3. Domain Knowledge Integration: The analytical procedure can incorporate physical principles and domain knowledge [115] through signal processing tools. With a better understanding of the physics behind the equipment and the expected failure signatures, engineers can develop signal-processing algorithms that result in more precise and efficient analysis procedures.

4. Robustness to Data Availability: When compared to AI-based techniques, signal processing techniques tend to be able to adapt to changes in data availability, quality, and labelling [116]. While AI models would need a lot of labelled data for training and validation. Signal processing methods may work well with smaller datasets or insufficient information.

5. Transparent Modeling Pipeline: Signal processing methods often use a straightforward processing pipeline with well-defined processing steps [117], which makes it easier to identify and debug problems that are developed during analysis. Methods based on artificial intelligence may provide more complex model frameworks and hyperparameter adjustments, which would result in complex pipelines.

### 3.5. Advantages of Preprocessing Based Methods

Preprocessing approaches improve raw vibration data by dealing with many challenges and preparing it for accurate analysis. These methods help to improve raw vibration data in a variety of ways.

1. Noise reduction: Vibration signals sometimes contain noise from a variety of sources, including electromagnetic interference, sensor faults, and also due to environmental conditions. Preprocessing techniques like averaging, filtering and wavelet denoising, etc., are used to minimize or remove noise and improve the quality of vibration data [118].

2. Baseline Correction: Vibration signals may show baseline offset [119], which makes it challenging to detect important patterns. To ensure that the analysis focuses on what is useful in the vibration data, preprocessing techniques like baseline correction and detrending are used to eliminate these unwanted trends.

3. Signal Segmentation: By using preprocessing techniques, it is possible to select specific events or conditions for study by splitting the raw vibration data into segments or windows [120]. The identification of failure signals, or modifications in machinery behaviour that might be hidden in continuous data streams is made easier by this segmentation.

4. Feature Extraction: Feature extraction algorithms are also used in preprocessing methods to convert unprocessed vibration data to a more useful representation. For further analysis and defect detection, these features record important parameters of vibration signals, such as frequency components, amplitude spectrum qualities and so on so forth.

5. Normalisation: When combining vibration data from several sensors or machinery units, preprocessing methods such as normalisation can guarantee uniformity by adjusting the data to a common range of reference.

6. Dimensionality reduction: Principal component analysis (PCA) and singular value decomposition (SVD) are two preprocessing methods that can reduce the dimensionality of the high dimensional vibration data set with different sensors or characteristics while maintaining useful information. This reduction reduces further analysis tasks, and improves computational efficiency [121].

### 3.6. Advantages of Artificial Intelligence Based Processing Methods

AI-based methods have advantages such as automatic feature learning, flexibility to complicated patterns, adaptability to unpredictability, end-to-end learning, scalability, and continuous learning. These advantages make AI-based approaches useful tools

for vibration-based condition monitoring, especially in applications with complicated fault patterns.

1. Automatic Feature Extraction: Artificial Intelligence algorithms, especially those using deep learning models, can automatically extract useful features [122] from raw vibration data, eliminating the need for complex feature engineering. This automatic feature learning ability can detect complex patterns and relationships in data that would be impossible to extract with standard signal processing approaches.
2. Adaptability to Complex Data Patterns: Artificial intelligence-based models can learn complex and nonlinear relationships in high-dimensional data [123], which allows them to discover minor fault signals and anomalies that traditional signal processing approaches might not recognise. Adjusting to complicated patterns allows artificial intelligence algorithms to identify and diagnose faults with high accuracy.
3. Robustness to Variability: Artificial intelligence models can be generalised effectively [124] over a wide range of operational situations, including speed, load and environmental factors. Since algorithms based on AI are robust to variability, they can be applied in real-world scenarios that require complex and dynamic machinery behaviour, an area in which traditional signal processing techniques could find it difficult to adjust.
4. End-to-end Learning: Without the requirement for intermediate feature extraction and selection processes, deep learning models have the ability to perform end-to-end learning [125], converting raw vibration signals to defect detection results. With this method, feature engineering may be conducted more effectively, with fewer dependencies on domain expertise, and with a simpler modelling pipeline.
5. Scalability: AI-based approaches can handle huge amounts of data and high-dimensional feature spaces, making them ideal for analysing data from many different sources. The ability to scale [126] allows AI models access to plenty of information and perform full machinery health evaluations. The ability to scale allows AI models access to plenty of information and perform full machinery health evaluations.
6. Continuous Learning and Adaptation: AI models may be constantly updated and enhanced with new data [127], allowing them to change in response to changing industrial situations and learn from previous experiences, improving performance over time.

In our recent paper [128], we deeply cover the important role of predictive maintenance for heavy machinery, the different needs of varying heavy machinery for vibration-based condition monitoring, the many benefits offered by vibration-based condition monitoring, the major obstacles faced in implementing this approach for heavy machinery and the major effect of breakdowns on operational costs and safety.

## 4. Summary

### 4.1. Signal Processing Based Methods

4.1.1. Time Domain Based Methods

Wu et al. [44] proposed the Winding Mechanical-Electromagnetic Coupling Model (MECM) for vibration-based transformer condition monitoring. MECM considers the relationship between stiffness, vibration, and electromagnetic forces. It uses distinct numerical techniques to calculate vibration responses with nonlinearity and electromechanical coupling. NEMC becomes more intense with vibration, especially in mechanical failures. A Nonlinearity-ElectroMechanical Coupling Coefficient (NEMCC) has been created to predict winding looseness trends to monitor winding mechanical conditions. Mean absolute precision error (MAPE) was used to detect degrees of winding looseness defects more accurately than traditional methods, creating effective winding mechanical condition monitoring conditions. Bhuiyan et al. [45] designed a cost-effective wireless vibration monitoring system specifically for remote machine vibration analysis in textile factories. The device helps in finding machinery that needs maintenance. The system uses a SW-420 vibration sensor on an Arduino UNO R3 to gather vibration data. It comprises two parts: a sender

unit and a receiver unit. The data are sent via Bluetooth from an SD card to a smartphone. The results showed that because low-quality devices are less durable, they vibrate more. Also, generators in a utility section vibrate more than boilers because boilers aren't made of mechanical parts. Cornel et al. [46] deal with two key elements: estimating pre-warning timeframes for probable damage using acoustic emission and standard vibration sensors and measuring the sensitivity of an acoustic-emission system in identifying scenarios that contribute to radial roller bearing failures. The mechanical condition of power transformer windings was determined using Operational Deflection Shapes (ODSs) by Shi et al. [48]. A calibration measuring system records signals related to excitation voltage and tank vibration. ODSs can be distinguished from one another using descriptive metrics such as Pearson Correlation Coefficient (CC) and Histogram of Oriented Gradient (HOG). According to the study, ODSs showed waveform changes and winding vibrations that mostly corresponded with power frequencies. Changes in operating conditions, such as voltage and current, affect the amplitude but not the fundamental shape of ODSs. Although it doesn't detect specific winding failures, this method provides a simple fault diagnosis method that is less affected by routine operations. A complete diagnosis requires more investigation. Raghavendra Pai et al. [49] described how to analyse complex vibration data from an industrial machine. It presents an efficient technique for simplifying examining a 100 HP Kirloskar motor's vibration-based condition monitoring. Maintenance records were examined for deviations from standard operating procedures. For 150 days, vibration readings were taken in the X, Y, and Z directions at both the front and back ends of the motor. The average delay duration was 28 minutes at first, with a maximum of 70 minutes. The wait time was decreased by 95% after introducing predictive maintenance, increasing productivity and machine efficiency. They concluded that systematic, step-by-step maintenance is important for successful predictive maintenance, as is proper device selection and sensor placement for accurate results during data post-processing. An improved method for identifying rolling element-bearing operating conditions that may cause premature failure was presented by Marticorena et al. [50]. This method measures the bearing cage's kinematics, which is determined by the traction forces created in the rolling elements' contacts with the races. The method is tested in a test rig that includes a cage IAS sensor, an AE transducer, and a piezoelectric accelerometer. In order to evaluate the lubrication regime, signal data are collected and preprocessed using angular resampling. Digital low-pass filtering, shaft rotation measurement, and peak and zero crossing detection were applied between the same pocket. Morgan et al. [47] looked at using vibration analysis to monitor the lubrication state in a toggle clamping system in real time. To identify vibrational patterns associated with lubricant degradation, they used structural response analysis, including traditional time-domain and frequency-domain analyses. In the initial phase, two common health indicators for vibration analysis were tested: the frequency centre (FC) health indicator in the frequency domain and the root mean square (RMS) health indicator in the time domain. The CLF-60TX plastic injection moulding machine was used for the experiments, and an IMC CRONOSflex data acquisition system with eight channels and a maximum sampling rate of 100 KS/s was used.

### 4.1.2. Frequency Domain Based Methods

Skowronek et al. [51] proposed a new approach for local damage detection by highlighting the role of background noise features in signal analysis. When using traditional damage detection techniques, they highlight the importance of probabilistic noise features like Gaussian or non-Gaussian distributions. In order to assess whether classical methods are appropriate, they proposed an intuitive methodology for detecting variance in signal representations. This method can be useful for those dealing with noisy signals, especially when non-Gaussian. The importance of defect size on vibration responses and acoustic emission (AE) in bearing outer races was examined by Nirwan et al. [52]. Vibration analysis proves the existence of a defect but not its size, demonstrating an increase in overall RMS velocity with rising RPM, load, and defect size. On the other hand, super harmonics

and defect frequencies are highlighted by acoustic emission analysis, where the dB value increases with RPM, load, and defect size. Mehamud et al. [54] used vibration sensors based on Triboelectric Nanogenerators (TENG) for industrial applications and were successfully fabricated. The spring-assisted TENG demonstrated improved performance and power output due to structural design based on computed natural frequency simulations. The designed sensor has an outstanding level of linearity and stability with a frequency detection range of 0–1200 Hz and an acceleration range of 0–120 $\frac{m}{s^2}$ .This sensor outperforms previous TENG-based vibration sensors with frequencies that are below 200 Hz. Based on FFT signal spectrum analysis, TENG-based vibration sensors show significant promise for measuring low-amplitude vibrations and identifying bearing defects, making them valuable for a wide range of frequency applications with excellent accuracy and the potential for self-powered sensor development. Gurusamy et al. [55] described the latest advances in magnetic flux-based condition monitoring of electrical machinery. It examines sensor technology, focusing on search coils and Hall sensors for internal flux sensing, as well as coil sensors with magnetic cores and fluxgate sensors for outward flux measurement. Magnetic flux may efficiently identify motor issues and provide current-based repairs. While some applications might require several sensors, lowering their number without dropping diagnostic accuracy requires more study. The design and implementation of a transmission tower monitoring system was discussed by Wang et al. [56]. Vibration analysis experiments were carried out to verify the gearbox towers' horizontal vibration responses—which were produced by striking the towers from either the X or Y direction. The force-generating device, wireless acceleration sensor, monitoring centre, and monitoring device make up the designed monitoring system. Through a 4G network, vibration data collected by the acceleration sensor is sent to the monitoring centre for study. The study shows that real-time measurements of a gearbox tower's vibration response may offer valuable information. Particularly, when force pulses are applied in the X and Y directions, the acceleration response can be used to build the frequency response function of the structural system of the gearbox tower. A triaxial accelerometer was used in the study to detect acceleration along three axes. The main objective of the study is on the natural frequency variation following settlement, which is an important parameter for tracking gearbox conditions. Hu et al. [57] gathered Vibration data from accelerometers and a speed sensor at a high sample rate in the study of wind turbine gearbox condition monitoring. Damage to the second-stage ring-gear tooth was found during an inspection. A study was performed using accelerometer data, which showed blurring effects caused by variable-speed operations, making diagnosing gear degradation difficult. A synchronous analysis technique with instantaneous shaft speed determination was used to address this. The second harmonic of the high-speed gear meshing was identified as a speed-tracking index by analysing the vibration signal using a proposed approach based on Short-Time Fourier Transform (STFT), which aligned well with the gearbox's high-speed shaft (HSS) reference speed. Priebe et al. [23] proposed a vibration-based approach to condition monitoring (CM) in applications that require parts subjected to wear and challenging operational circumstances, such as cutting discs in tunnel boring machines. The study focuses on the feasibility of using vibration-based approaches to monitor TBM cutting discs. Data for performing experiments was gathered via an RTD equipped with a load cell and an accelerometer, resulting in time-series data at a sampling frequency of 1 kHz. High-resolution images of the damage are collected and measured to properly measure damage size, alongside a scale for reference. Peters et al. [58] validates the feasibility of embedding MEMS sensor systems onto gears, allowing important vibration properties to be measured for condition monitoring and wear prediction. These systems are cheap and easy to combine with existing sensors and microcontrollers. The frequency peaks found in the spectrum have been attributed to gear meshing mechanics, which ensures measurement precision. The integrated MEMS sensors outperform reference sensors on the bearing block, providing a higher signal-to-noise ratio. Peeters et al. [59] assumed that rotating component defects, such as bearings and gears, generate second-order cyclostationary content in signals. It chooses an appropriate sparsity measure to optimise the

sparsity of the squared envelope spectrum. The approach is validated using simulated cases and NASA's Prognostics data pool. This method uses blind filtering and the envelope spectrum to detect defects in vibration signals with a second-order cyclostationary signature. It performs better than standard blind deconvolution methods when fault signals have important impulsive content. Simulations and experiments both verify the method's ability to find cyclostationary sources. Aburakhia et al. [5] described a hybrid technique for vibration-based rolling bearing condition monitoring and fault diagnostics. The method was tested on many bearing datasets and found to be more accurate and faster than earlier approaches. It combines wavelet packet transform and Fourier analysis to break down short segments of vibration signals into component waveforms, allowing spectrum information to be extracted. The proposed approach efficiently computes fault-sensitive features by utilising the important frequencies in fundamental waveforms. The application of acoustic signal analysis for detecting possible faults in induction motors was discussed by Rajapaksha et al. [39]. They suggest the importance of acoustic signals in fault diagnostics, exposing their potential. While time domain features are important for bearing failure identification, they lack information about frequency components and do not pinpoint the location of the problem. Peak value, RMS, crest factor, skewness, and kurtosis are all recommended statistical variables for this kind of analysis. Frequency domain spectrum analysis was used to locate the problem, giving an improved method for fault identification in induction motors.

### 4.1.3. Time Frequency Domain Based Methods

The triboelectric nanogenerator is a very sensitive self-powered vibration sensor presented by Zhao et al. [61] for equipment condition monitoring. The VS-TENG detects vibrations across a wide frequency range (1 to 2000 Hz) and can detect low-amplitude vibrations. This study explores its performance under various vibration situations in detail. With a detection accuracy of 99.78%, the triboelectric nanogenerator is ideal for monitoring mechanical gear systems. Mystkowski et al. [60] designed a relatively inexpensive vibration-based machine monitoring system focusing on an electronic prototype. The system used a four-channel signal converter built on an AVR microcontroller ATmega128 with a 10-bit ADC. While assembler software was more efficient than C language software, it had limited frequency resolution. Real-world measurements on a hay-handling machine show that the system can perform simple diagnostics, primarily in the time domain, but more advanced frequency techniques are required for in-depth analysis. Michalak et al. [33] proposed a signal model for the vibrations of an industrial vibrating sieving screen suspension. The model was constructed by analysing actual machine signals from the mining sector. An autoregressive model, including internal and external signals, is used to model impulse response. Morgan et al. [21] created a data-driven denoising system for separating vibrational fault signs and background noise. Their proposed model combined the singular value decomposition method with continuous wavelet transform. It takes a noisy time domain signal as an input, processes it with wavelet transform and singular value decomposition in order to remove noise, and then uses inverse wavelet transform and sigmoid thresholding to provide denoised signals.

### 4.1.4. Preprocessing Based Methods

Yin et al. [15] introduced a new three-dimensional data compression technique to effectively reduce data size without compromising bearing defects detection accuracy. By binarising vibration data, a new compression dimension was added, implementing the advantage of the bit cost of individual sample points and other dimensions involving sample points and frequency components. In terms of compression and diagnostic performance, it outperformed five previous approaches. The proposed method considerably improves data compression, making it useful for managing large amounts of data. More importantly, it shows high generalisation for multiple data patterns, not just vibration data. Ghazali et al. [62] presented a real-time vibration-based anomaly inspection for

drones, specifically focusing on drone arm and propeller components. The capacity of MEMS vibration sensors, such as the accelerometers ADXL335 and ADXL345, to identify issues from raw acceleration data was evaluated. According to the study, the ADXL335 accelerometer performed better than the others and was selected for Arduino DUE integration. This enables users to monitor the drone's health and malfunctions via a mobile application. Sharma et al. [63] measured vibration signals by sensors mounted on a steel truss bridge. These signals have been studied in the frequency domain (FFT) and the time-frequency domain (spectrogram and Stockwell transform). After noise was found in the signals, a series of low-pass filters were applied, coupled with Hilbert transform (HT) and frequency resolution-enhancing methods, to eliminate noise and frequencies higher than 30 Hz. Various low-pass filtering techniques, such as the variational filter, Savitzky–Golay (SG) filter, and lowpass sparse banded (AB) filter, were used to refine the vibration signals. Due to noise, the FFT analysis showed no ideal frequency peaks near the required modal frequencies. MEMS dual-channel accelerometers were properly mounted on the bridge to capture the vibrational response data. Bhowmik et al. [2] presented an innovative approach that combines filtering, enhancement, fault detection, and modal identification into a unified framework. The main goal is to create a real-time condition monitoring method for mechanical vibrating systems that uses only a single sensor-based output algorithm. The study uses the principles of first-order Eigen perturbation to develop an online filtering technique based on recursive singular spectrum analysis (RSSA). The algorithm uses the first-order Eigen perturbation (FOEP) methodology's principles to continuously update the Eigenspace at each time step while working in real-time. This unified framework provides a comprehensive solution for efficiently and effectively monitoring the condition of mechanical vibrating systems. Rafiq et al. [36] introduced NAMEMD, an improved Multivariate Empirical Mode Decomposition (MEMD) variant. NAMEMD is designed to improve the multivariate Intrinsic Mode Functions (IMFs) decomposition by adding uncorrelated white Gaussian noise into individual channels under specific conditions. The aim of adding Gaussian white noise to the decomposed multivariate IMFs is to reduce mode mixing problems. The method requires carefully choosing the parameters to introduce white noise to the original multivariate signal in a different channel. This substantially decreases modal aliasing in MEMD, which reduces the impact between different IMFs in each channel. The study uses synthetic signals and actual experimental vibration data from wind turbines to confirm the efficiency of this methodology. This technique is especially useful for handling multi-channel vibration signals with different speeds and loads. Fernando et al. [66] proposed a vibration-based technique for detecting bridge damage. It uses experimental modal identification to determine modal properties and validate a numerical model in an undamaged state. In two case studies, damage is identified with the help of the comparison between the experimental and numerically determined modal parameters. Global damage index (DI) parameters are created and evaluated to evaluate damage based on natural frequencies, mode shapes, and modal damping ratios. The analysis effectively finds two vibration modes in the bridges. It highlights the need to identify stable damping ratios and very sensitive sensors in sufficient numbers for accurate damage location. Patil et al. [67] combined an asperity-based acoustic emission model for rolling element bearings with a multi-body dynamic model to present a novel method for acoustic emission modelling. The model produced was used to investigate the effect of load, speed, and radial clearance on acoustic emission. The results show that acoustic emission increases with larger loads increases up to a particular speed threshold, and then drops. To provide a comprehensive understanding of the acoustic emission behaviour, the model provided in this work basically integrates the multi-body dynamic (MBD) and REB AE models. To evaluate the structural condition changes caused by bridge expansion, Lin et al. [68] highlight the utilisation of acceleration and strain data obtained from a short-span concrete culvert bridge in New South Wales, Australia. To show the changes in structural condition, it compares two identification algorithms: frequency-domain decomposition (FDD) and frequency spatial decomposition (FSDD). This analysis effectively determines the decline

in modal frequency with the bridge extension event. The results show that the bridge's natural frequency dropped from roughly 16 to 12 Hz due to the additional deck mass caused by the bridge enlargement. Zonzini et al. [65] described a sensor network intended for structural health monitoring (SHM) applications, to extract synchronised modal parameters from structures. Microelectromechanical systems accelerometers and low-cost piezoelectric devices were used in the network. This combination of sensor technologies allows for low-frequency range effective structure health monitoring. The sensor nodes in the network architecture are synchronised and use triaxial MEMS accelerometers in addition to PZT transducers. These nodes are connected to a network interface or gateway (GW), allowing real-time data transmission and analysis on a PC. Synchronisation is ensured by each sensor node acquiring data simultaneously. They used the AR+Noise technique to handle noise levels in real data, increasing its applicability in various situations. The research results show that inexpensive PZT sensors, used alone or in conjunction with traditional MEMS accelerometers, can effectively estimate the modal parameters of structures going through flexible vibrations, providing a less intrusive method of SHM. Industrial ICP accelerometers were permanently placed by Krot et al. [64] on the gearbox's input shaft bearing supports. They were recording vibration data at a sample frequency of 2 kHz. The torsional load measurements from the motor shaft and angular clearance checks in the gearbox were also collected. The torque and vibration tests showed that the dynamic response of the shaft varied with rotational direction, with the highest vibration happening where opposite forces acted on the shaft. They concluded that monitoring the shaft's natural frequency and analysing spectrum amplitudes as well as phases at higher harmonics could diagnose problems such as radial clearance and fastening bolt loosening in bearing supports. Natural frequency changes correspond to bearing support stiffness and joint opening, indicating wear or larger gaps.

### 4.2. Artificial Intelligence Based Methods

Machine Learning Based Methods

Nowakowski et al. [30] described a tram gearbox diagnostic technique that uses noise data obtained during tram operation. Under the same driving conditions, psychoacoustic measures, particularly loudness and fluctuation strength, show considerable differences in the acoustic properties of the axle between healthy and problematic gearboxes. The study aims to use Empirical Mode Decomposition (EMD) for signal analysis and combines it with a decision tree method to find defects in the tram's gearbox. The approach was highly effective, with only 3% of unnecessary fixes and no missed damage. Using a single trackside microphone results in a 33% decrease in the cost of measurement equipment. To reduce background noise in bearing signals, Li et al. [31] applied symbolic dynamic filtering, which is especially well-suited for entropy analysis. Using fast spectral kurtosis, a new metric called mean spectral kurtosis has been introduced to measure the information content related to defects. They used SVM for pattern identification, specifically defect detection, and the OSE approach for feature extraction. Data collection, partitioning based on specific methods, symbolisation via symbolic dynamic filtering, feature extraction based on MOTE, and SVM training for classification were all part of the process; 75% of the data were used for training and 25% for testing the model's performance. They classified the faults into three types: inner race faults, outer race faults and rolling element faults. Gildish et al. [19] evaluated three regression approaches: Ridge, support vectors, and deep learning regression and determined to provide accurate predictions of operational conditions. The experiments were carried out on ten datasets, with half of the data used for training and the other half for testing. The Support Vector Regressor performs slightly better. Vibration signals have been found to transfer useful information about gearbox working states, such as speed and torque, and they were found to be useful for condition monitoring. Wang et al. [13] present a new vibration signal-based method that uses computer vision to monitor rail track infrastructure's non-destructive health. This approach takes raw vibration signals and turns them into grayscale images directly, unlike tradi-

tional approaches requiring noise elimination and feature reduction. Instead of using one-dimensional arrays for signal processing, it uses multi-dimensional image feature vectors. The FAST-Unoriented-SIFT method was used to extract many features from vibration signals, and a VBoW model with optimal keywords was given for defining and detecting grayscale image features. On the testing set, the model achieved a remarkable 96.7% recognition rate. This approach, when compared to traditional and deep learning classification algorithms, is better suited for recognising important time-varying and random vibration signals, indicating great potential in the practical monitoring of structural health applications. They classified the data into three classes including normal, minor damage and severe damage instances. Kannan et al. [9] discuss concerns of integrity in multisensor condition monitoring systems where sensor distortions may affect condition determination. They propose an approach that dynamically assigns each sensor's weight based on signal quality, efficiently using both heterogeneous and homogeneous sensor data. After that, they applied a classifier for every kind of sensor to obtain an output that identified faults for every sensor. After that, every decision was fused and weighted based on the normalised signal integrity score to produce a more accurate classification forecast. According to validation on various sensors, the proposed approach improved the accuracy and reliability for carrying out health condition evaluations, especially when individual sensors have limitations. To determine important features for defect identification and classification in rotating machine monitoring programmes, Brusa et al. [73] investigated the application of Shapley additive explanation (SHAP). They focused on medium-sized industrial bearings that were available from Politecnico di Torino's Mechanical Engineering Laboratory. They use SHAP as the feature selection criterion for support vector machines (SVM) and k-nearest neighbours (KNN) models, achieving accuracies of more than 98.5%. They classified the features into three classes: normal, inner race faults and outer race faults. SUN et al. [82] proposed a vibration signal-based fault diagnostic approach, where they used variational mode decomposition (VMD) for signal preprocessing, which performed better than empirical mode decomposition. A two-stage feature selection method combining Fisher discrimination and Relief was proposed, resulting in a highly accurate fault detection by considerably decreasing feature dimensions from 240 to 35. The approach's effectiveness has been verified by comparative studies, which identify two switching processes with 100% and 96.57% accuracy, accordingly. They classified the records into eight classes. Tsunashima et al. [85] developed a planetary gearbox's electrical current signature-based status monitoring system. Residual signals were processed to obtain weighted multi-scale fluctuation-based dispersion entropy features, which were then fed into Random Forest and KNN classifiers for classification. Results showed that, in this case, wtMFDE features extracted from residual signals provided accurate results. Because they accurately describe the dynamic behaviour associated with gearbox wear and collect important data from the residual signals, wtMFDE features are expected to provide correct results and improve identification performance. They classified the faults into three types: Longitudinal level, alignment level and cross level. Sharma et al. [37] developed an electrical current signature-based planetary gearbox condition monitoring system. Sensors and accelerometers were used to collect data, and residual signals were processed to extract weighted multi-scale fluctuation-based dispersion entropy (wtMFDE) features. Two classifiers were used for classification: K-Nearest Neighbors (KNN) and Random forest. The results showed that the wtMFDE features extracted from residual signals achieved acceptable accuracy when used for planetary gearbox condition monitoring. Sun et al. [74] collected vibration signals from a steel truss bridge by using sensors. Their analysis includes frequency domain analysis using FFT and time-frequency domain analysis using spectrograms and the Stockwell transform. A series of low pass filters combined with HT and frequency resolution improvement were used to remove frequencies above 30 Hz. They selected useful features using ReliefF and classified them using a support vector machine. They classified the instances in eight classes. Patange et al. [76] used machine learning (ML) and classical tool condition monitoring analysis to classify tool defects during stainless steel

turning. Vibration signals detect various stages of tool failure, and a decision tree based on many binary rules is created. The Random forest classifier has a maximum accuracy of 92.66%, making tree-based classifiers excellent for tool failure detection. The study provides the potential for onboard condition monitoring and proposes a further future investigation of low-cost, open-source software integration. They classified the signals into six classes: Good, Notching, Worn nose, Worn flank, Broken tip and Cratering. It fails to recognise that different parameters related to machines have an impact on tool conditions and that more varied data are required to create reliable classification models. Mukherjee et al. [83] introduced a new edge-based ubiquitous domain adaptation (UDA) method for lightweight machine condition monitoring. UDA enables the system to determine the operational conditions of one type of motorised equipment, even when training on completely different equipment. The proposed UDA method ensures domain invariance in machine status monitoring by using signal expansion and principal component analysis (PCA). It uses data from all three axes of the vibration sensor to capture blender motor vibrations. This method effectively reduces noise artefacts and device-specific vibration signatures using basic time-frequency domain signal operations and a data-driven ensemble of classifiers. To identify the operating states of the target domain, the experimental setup involves using a machine state identification classifier with vibration data from an air-cooled electric blender motor (source domain). The data are divided into training and testing sets in the source domain, with a 70:30 ratio, and validated against known data from different machines. They labelled the data in two classes: air and load. Balachandar et al. [38] used machine learning algorithms, such as decision trees, logistic model trees, Hoeffding, and random forests, to analyse vibrations for tool condition monitoring in friction stir welding (FSW). A piezoelectric accelerometer that was fixed to the tool head was used to obtain vibration signals in three distinct tool conditions: good, air gap, and broken. A data acquisition system and NI LabVIEW were used to digitise the signals. From the raw vibration signals, statistical features like mean, median, mode, standard deviation, and others have been extracted. A decision tree was used for feature selection. The instrument was in excellent condition when the experiment started, and signals were recorded. Next, each fault was simulated, and corresponding signals were recorded. Among the four classifiers used, the random forest produced the best classification accuracy of 93.51%. Using a piezoelectric accelerometer, Pranesh et al. [77] collected vibration signals from a brake system under several failure scenarios. Compared to SMO and MLP, a logistic algorithm produced the best classification accuracy, at 85.52%. They focused on brake condition monitoring based on statistical features extracted from vibration signals. All three algorithms analysed and classified thirteen features. The study evaluated six faults and good conditions by analysing raw vibration signals. The fault conditions were air in the brake fluid, brake oil spill, brake pad wear even inner and outer, brake bad wear even inner, uneven brake pad wear inner and uneven brake pad wear inner and outer. The logistic method outperformed MLP and SMO, with a maximum classification accuracy of 87.45% and 82% and 83%, respectively. Patange et al. [78] considered both perfect and problematic configurations of four insert tools. Statistical features such as Kurtosis, Standard error, and others were extracted using an event-driven algorithm. They proposed a machine learning-based method that uses six supervised tree-based algorithms for fault classification to determine the condition of complex tool inserts. They classified faults into four types: Defect, wear at the flank, wear at the nose and notch wear. Their evaluation showed the most promising classifier as the best-first tree classifier that achieved 97% accuracy, confirming the model's best training. Harish et al. [79] started to work with a statistical theory and then used it to develop an interpreted prototype neural network. It has been proven through data from experiments that there is an important correlation between the weights in the hidden nodes of the hidden layer and the informative frequency bands and fault characteristic frequencies. This strategy aims to make machine condition monitoring and classification easier, by applying artificial intelligence to diagnose automobile brake issues by analysing vibration data. Seven different fault scenarios were simulated using

a hydraulic brake setup. A piezoelectric accelerometer was used to record vibration signals under both ideal and defective braking situations. Twelve statistical features, such as range, skewness, kurtosis, and others, were extracted from these signals to identify features that would be suitable for the study. An attribute evaluator was used to select features. The hydraulic brake system of a commercial passenger LMV served as the test configuration. A piezoelectric uniaxial accelerometer with 10 mV/g sensitivity was used to obtain vibration signals under various fault circumstances. In classifying brake failures, the Logit boost algorithm achieved a maximum accuracy of 88.22%. Balachandar et al. [80] used the best first tree technique to extract statistical features from raw vibration signals to classify faults related to FSW tools. To maximise performance, features were selected, and the best first tree that had been post-pruned showed more accuracy at 1800 rpm, while the pre-pruned tree performed better at 1400 rpm. They classified the instances into three classes: good, bad and broken. As a result, it is advised that the post-pruned best first tree be used as a reliable model to forecast the FSW tool condition. Lu et al. [86] proposed a data-driven early-warning approach to possible vibration issues and a unique framework for monitoring mill vibrations in a cold rolling mill. They reviewed support vector regression (SVR), neural network-based (NN-based), and XGBoost models for forecasting vibration acceleration. The ability of the proposed approach to detect mill vibration changes under varying roll settings, as well as the superior performance; the XGBoost model performed best. They treated this problem as a regression problem. Lipinski et al. [81] focuses on using decision trees to classify data that represent the condition of a gearbox. They were processing a 15-dimensional vector to determine the gearbox's condition, and they could classify different gearbox conditions correctly. They used these 15 features collected from vibration data and processed through spectral analysis to build a model to identify different states or faults within the gearbox. They classified the condition of a gearbox as either good condition or bad condition. Joshuva et al. [75] determine possible defects in wind turbine blades by analysing their condition. The tests were conducted on a 50 W, 12 V variable wind turbine on a fixed steel stand in front of an open circuit wind tunnel outlet. Wind speeds ranging from 5 to 15 metres per second were used to simulate various environmental situations. A piezoelectric accelerometer was positioned near the wind turbine hub and connected to a DAQ system via a wire for collecting vibration signals. The turbine was initially fault-free, and vibration signals were recorded in NI LabVIEW. Histogram features were computed from these signals and selected using a J48 decision tree method. With an average computation time of 0.07 seconds. The locally weighted learning model produced a 93.83% fault classification accuracy. Following that, different flaws, such as blade bend, crack, erosion, loose contact at the hub, and blade pitch angle twist, were simulated one at a time on individual blades while keeping other blades in good condition. Each fault state generated various vibration signatures, that were captured and analysed to determine fault diagnosis and characterisation. Gómez et al. [84] collected vibration response data for wheelsets from Dannobat Railway Systems. Six uniaxial acceleration sensors have been carefully positioned on each wheelset to record vibrations in both the radial and axial directions. Their method combines vibration analysis with wavelet packet transform (WPT) energy and a support vector machine (SVM) as a diagnosis model. They classified the signals as either corresponding to cracked or healthy status.

### 4.3. Unsupervised Machine Learning Methods

Mazzoleni et al. [34] provides a method for diagnosis that combines vibration data envelope analysis, qualitative machine performance data, and anomaly detection algorithms for mechanical component condition monitoring, including gearboxes. All of the fault indications are combined using a fuzzy inference technique. The widespread use of unlabeled data in machinery with vibration measurements motivates anomaly detection and envelope analysis. Using two preprocessing steps—Principal Component Analysis (PCA) to reduce dimensionality and select key feature combinations. Using Gaussian Mixture Model (GMM) to find outliers related to nominal data distribution—the proposed method

improves on the traditional one-class support vector machine design. Hendrickx et al. [32] discussed an unsupervised anomaly detection framework for machine fleet condition monitoring. Compared with traditional techniques, this framework makes comparisons in real time within the fleet and does not rely on historical data. Because it can detect any deviations in machine behaviour, it eliminates the requirement of prior knowledge of all possible machine problems, making it a more flexible method. Nie et al. [87] proposed a fixed moving principal component analysis (FMPCA), a data-driven damage detection technique, to track bridge damage's occurrence and operational state. Damage indices were continuously computed using a fixed moving window and principal components. The effectiveness of FMPCA in analysing dynamic vibration data for damage identification has been shown through simulation and experiments.

Deep Learning Based Methods

To develop intelligent fault diagnosis models, Espinoza Sepúlveda et al. [89] combined machine vibration responses with known machine faults. A supervised smart fault diagnosis model has been created specifically using measured vibration responses from a rotating rig with various simulated rotor faults. The main goal of this study was to develop a smart vibration-based machine learning model for fault diagnosis under particular operating conditions and then assess how well it could adapt to other scenarios. The data used in this study are pre-existing records gathered on an experimental laboratory rig. Four uniaxial accelerometers were used to collect vibration data at a sampling rate of 10 KHz. The data were classified into two classes: Healthy and Faulty conditions, with four scalar features extracted from each data sample. These features included the root mean square (RMS) and statistical parameters like variance, skewness, and kurtosis. A Multi-Layer Perceptron (MLP) was used for pattern recognition and classification of the acquired vibration data. For the development and evaluation of the model, the dataset was divided into training, validation and testing with a ratio of 70%, 15% and 15%, respectively. Demircan et al. [92] used a three-axis accelerometer and an electrical current sensor to record vibration and current signals, respectively. These signals are captured in a variety of conditions, including healthy and worn impellers, cavitation caused by throttling the inlet valve, and mechanical looseness caused by loosening a screw. At motor speeds ranging from 250 to 800 rpm, vibration signals have been collected in radial horizontal, radial vertical, and axial planes (x, y, and z axes). They explore classifiers based on features derived from many representations of a vibration signal: time, frequency, and time-frequency domains. They collected 30 features from these signals, including 16 Shannon entropy features and six wavelet variance estimates. They then trained different classifiers. The neural network was the most accurate, with a test accuracy of 99.0%. A hardware-based neural network for planetary gearbox status monitoring was presented by Dabrowski et al. [22], focusing on non-stationary situations and speed constraints. Important features include high performance (125 kHz), dependability, and low power consumption, all of which can be achieved on a single FPGA. The system has been incorporated into LabVIEW for versatility and low data transmission, with the potential for parallel operation with monitoring systems. They achieved 94% accuracy in identifying known technical faults and 86% accuracy in identifying unknown technical faults. The FPGA beats the PC in complex algorithms but falls below simpler ones. They classified the signals broadly into two classes: correct and incorrect. At the low level, they also sub-classified each class into several sub-classes like correct, correct loading, correct varying load and so on and so forth. To distinguish between different degrees of gearbox wear, Elvira-Ortiz et al. [18] presents seven entropy-related features. They take time-domain signals and directly extract statistical features from them, without converting them to another domain. In nonlinear data, entropy features work well to identify dynamic behaviour. They combined three techniques: entropy features, linear discriminant analysis, and artificial neural networks to improve machine learning performance for wear degree identification in gears. Their solution performed better when comparing the entropy-based approach with one with statistical features. The main contribution of this work is the com-

bination of ANNs, LDA, and entropy features to develop a machine-learning method for the identification and categorisation of various gearbox wear levels. They have instanced for four different types of situations, including healthy gear, gear with 25%, 50% and 75% uniform wear. Their proposed model achieved an accuracy of 99.7% on training data and 99.7% on testing data. Wang et al. [27] started with a statistical theory and then used it to develop an interpreted prototype neural network. It has been proven through data from experiments that there is a substantial correlation between the weights in the hidden nodes of the hidden layer and the informative frequency bands and fault characteristic frequencies. This strategy aims to make machine condition monitoring and classification easier. Zonzini et al. [88] looked into the effect of data compression on machine learning-based damage identification in structural health monitoring. The study compresses the Z24 bridge dataset using the MRAK-CS technique and uses the compressed data to train low-complexity neural networks tasked with anomaly detection. They explored several important topics, such as the impact of compression ratio on detection performance, how network complexity affects classifier performance (ANN and OCCNN), how to implement this practically on devices with limited resources, the benefits of incorporating temperature data in network inputs, and how noise floors affect the classification scores of MEMS-based accelerometers. Their results showed that the OCCNN architecture, which has a compression ratio of 6 and eight neurons per layer, shows a small decrease in performance when compared to more complicated networks that use uncompressed data. They classified instances into two classes: healthy and damaged. This is especially interesting. The compression stage reduces the risk of network congestion, and the low computational cost is per the capabilities of inexpensive microcontrollers. Vibration data are collected from many components by Sepulveda et al. [90] in an experimental rotating rig, including steel shafts coupled by a rigid coupling and supported by four ball bearings. One shaft is connected to a three-phase electric motor via a flexible coupling and holds two balancing discs, while the other holds one. The inherent frequencies of the rig have been measured and correlated with specific mode shapes, making it easier to identify vibration issues. The data collected from accelerometers at various bearing cases provides information into the rig's health at various rotor speeds, including healthy and faulty states. At every bearing housing, these accelerometers with a sensitivity of 100 mV/g and a frequency range of up to 10 kHz have been carefully placed at 45-degree angles from vertical and horizontal directions. They have captured six different statuses including healthy, misalignment, looseness, bow, shaft bow and rub. They fed features including root mean square, variance, skewness, and kurtosis, extracted from time domain signal to multi-layer perception model fault identification.

Koutsoupakis et al. [42] developed and evaluated a damage identification and condition monitoring (CM) method based on Convolutional Neural Networks (CNNs) for a two-stage gearbox. Training on simulated data, the CM-CNN shows the model's stability and dependability in predicting experimental data under various conditions, including varying rotational speeds. They evaluated their model performance on accuracy, and they set the RPM to 612, 911 and 1210. For RPM of 612, they achieved an accuracy of 98.7%, for RPM of 911 they achieved an accuracy score of 91.3%. While they achieved an accuracy score of 81.3% for 1210 RPM. In their experiments, they have records from three statuses: healthy, damage 1 and damage 2. Ong et al. [1] describe an intelligent gearbox failure diagnostic system that uses Deep Convolutional Neural Networks (DCNN). The results demonstrate the DCNN framework's superiority over standard models (DT, RF, and SVM), with excellent average precision, sensitivity, specificity, and accuracy levels. The ability of DCNN to automatically extract features from data without any human involvement is interesting. This could include investigating multi-source information integration, data augmentation for small datasets, and DCNN enhancement via metaheuristic algorithms. They performed their experiments on data from three classes: normal, chipped tooth and worn. Inturi et al. [91] explore the use of the Hurst exponent in analysing aperiodic and non-stationary data that indicates scale invariance and self-similarity. Raw vibration and acoustic signals are collected to evaluate thirteen health stages of a multi-stage gearbox

including one healthy, four inner race faults, four outer race faults and four cracked tooth stages. The Hurst exponents are calculated using three methods: generalised Hurst exponent, rescale range statistical (R/S) analysis and dispersion analysis. When applied to the extracted features from vibration and acoustic signals collected from the multi-stage gearbox, deep learning classifiers outperform traditional machine learning classifiers such as decision trees and support vector machines, showing higher classification accuracies. Chesnes et al. [28] developed and evaluated two vibration-based techniques for detecting reciprocating compressors' early valve seat wear. It involves using time-frequency analysis to extract a region of interest (ROI) and train a convolutional neural network (CNN). Using texture and shape image statistics for feature extraction, a discriminant classifier trained on the same ROIs is compared to the deep learning approach. In CNN classification, both methods provide over 90% accuracy. The short-time Fourier transform (STFT) and decomposition of vibration signals into compression cycles are examples of signal processing techniques. Traditional statistical techniques (linear and quadratic discriminant) and a method based on deep learning that trains a CNN directly on ROIs are the two methodologies used for classification in their work. They classified the faults sometimes into two classes, and sometimes into three classes, but did not mention their categories. Amin et al. [4] applied vibration analysis and deep learning to identify and classify low-speed shaft wind turbine gearbox defects. They performed simulations using a range of realistic wind loads and transient forces. Both cyclostationary-based CNN and kurtogram-based CNN obtained above 80% accuracy in fault classification, particularly in detecting fault areas, according to the data. The Shannon Spectral Entropy (SSE) was used by Civera et al. [16] to look into the use of Instantaneous Spectral Entropy (ISE) as a time-dependent damage indicator for wind turbine gearbox vibration data. ISE is extracted using the Continuous Wavelet Transform (CWT), and a sensitivity analysis on Generalised Morse Wavelet (GMW) parameters is then carried out. A moving mean of ISE is used to improve stability. It detects bearing faults in experimental data successfully, with some false alarms, but real condition changes result in sustained deviations from the baseline. They focused on vibration response analysis using an ensemble of Convolutional Neural Networks (CNNs). They evolve these CNNs by feeding them frequency responses with distinct purposes. They use the Dempster-Shafer theory to combine the CNN outputs effectively. They use one actuator and two sensors to collect vibrational response data from turbine blades. The ensemble framework significantly improves the classification accuracy by using an improved Dempster-Shafer method to combine CNN outputs, increasing it from 94.05% to 96.28%. Amin et al. [94] proposed a machine-learning framework for vibration-based wind turbine gearbox monitoring. To convert vibration signals into images (Kurtogram and cyclostationary), they use signal preprocessing techniques such as the envelope criterion, cepstrum editing, spectral kurtosis, and cyclostationarity. Then, Convolutional Neural Networks (CNNs) are trained to detect defects; 70% of the data were used for training and validation, whereas 30% were used to test the model's performance. The paper provides two CNN models, one based on cyclostationary data and the other on kurtogram data which achieved an accuracy rate of 80%. They classified signals in one healthy and six damaged conditions. The important role of data preparation for vibration-based condition monitoring with CNN models was the focus of the work conducted by Yaghoubi Nasrabadi et al. [93]. The experiment used generative functions for data augmentation and a LASSO-based technique for data selection and reduction. Data selection dramatically lowers computational costs and improves classification performance. Also, the efficiency of generative functions is established. The results show that combining data selection and data augmentation can improve classification performance while also decreasing simulation time, providing important knowledge about appropriate data preparation for CNN models in condition monitoring. Taking an experimental lift door system as a test case, Koutsoupakis et al. [12] present an AI-based condition monitoring approach for identifying and detecting damage in mechanical systems. It shows how well an optimised Multibody Dynamics (MBD) model can simulate the system's behaviour.

Without additional optimisation, the MBD models for faulty states, which are based on the optimal model, accurately simulate damage. Even for minor damage like wheel wear, the Condition Monitoring Convolutional Neural Network (CM-CNN) achieves an average accuracy of 92.3% when trained using only simulation data. This highlights the importance of using simulations to address data shortages in real-world applications. The CNN ensemble generalises well to new datasets, reducing overfitting problems. Furthermore, a simple CNN framework with few parameters allows prediction generalisation by reducing differences between simulation and actual measurements. They classified signals among one healthy condition and two damaged conditions. Naveen Venkatesh et al. [95] discussed a method for monitoring single-point cutting tool conditions by applying transfer learning techniques. The vibration signals of the cutting tool have been collected and processed, and they were used as inputs for deep learning algorithms. Four pre-trained networks (Vgg-16, AlexNet, ResNet-50, and GoogleNet) were evaluated for the condition monitoring task. Once the turning action was stabilised, vibration signals were recorded using a piezoelectric accelerometer. After processing the data to remove noise and amplify them, the analogue to digital conversion was carried out to generate digital signals, and then vibration plots were drawn. These signals were then classified as good, tblent1, tblent1, and tiploose. Murgia et al. [17] examined how well data-driven SCADA-based condition monitoring works to identify faults in wind turbines. The approach is considered weakly supervised, based on models of normal wind turbine behaviour trained on datasets that do not contain information on component repairs. They considered controlled experiments with faulty and non-faulty turbines to show the suitability of a threshold-based alarm model for identifying drive-train failures. They used a three-year dataset for their experiments; the first two years were used for training, and the remaining data were used for validation and testing. In addition, they recommended using a threshold-based approach for diagnosis compared to prognosis, highlighting that this approach is more appropriate for fault detection and condition monitoring than for predictive maintenance. They used weakly instantiated convolutional neural networks for fault identification.

Vos et al. [40] proposed a novel data-driven architecture for semi-supervised anomaly detection in vibration signals, with an emphasis on gear wear and bearing defects. The combinations of the LSTM-regressor and one-class SVM classifier handle certain fault types. While the LSTM-OCSVM model is a useful tool for detecting gear defects, it may not be able to detect small changes resulting from bearing faults. The two-step LSTM2-OCSVM model (Architecture 2) has been proposed as a solution for this drawback, improving the detection of increasing bearing defects. They classified signals as either healthy or anomalous. Simple one-class SVM outlier detection based on statistical features works better. Afridi et al. [20] focused on developing a defect prognostic system for rolling element bearings, which are critical components in industrial systems and have major fault rates. The proposed model performs better than other models based on time domain, frequency domain, and time-frequency domain signal analysis. Vibration signal analysis is a popular and effective method for predicting the operation of rotating machinery. The study shows that eliminating noise in the frequency domain needs domain knowledge and may result in losing valuable information points the model requires for pattern analysis. They considered inner race and outer race faults for their experiments. They used LSTM network architecture for fault identification in bearings. With a focus on anomaly identification, Ahmad et al. [96] provides an auto-encoder model-based method for rotating machine condition monitoring. The technique uses normal vibration signals to learn the healthy status of a spinning machine, and it uses a threshold-based method to identify anomalies based on the reconstruction inaccuracy of unseen data. In particular, this method does not require feature engineering because it extracts important features from unprocessed vibration signals. Using two datasets of rotating machinery, an impressive average F1-score of 99.6% was obtained. The study provides an unsupervised method for automated feature extraction from unprocessed vibration data and shows that LSTM-based auto-encoders can accurately diagnose faults. The dataset was partitioned into training, two validation

sets, and a test set, with overfitting and gradient concerns addressed. LSTM-AE latent representations can be used instead of manual feature extraction, which requires prior domain and signal processing knowledge. Feng et al. [97] describe a novel gear prognostic scheme for predicting Remaining Useful Life (RUL) in the context of gear wear progression. The approach uses a new gear wear monitoring indicator based on Wasserstein distance and cyclic correntropy bispectral analysis. When it comes to precisely tracking and monitoring the progression of gear wear, this monitoring indicator works better than traditional indicators. It is combined into a Gated Recurrent Unit (GA-GRU) network that has been optimised using the Genetic Algorithm (GA) to fine-tune the hyperparameters of the GRU. This combination provides accurate RUL prediction for the gearbox. For the purpose of diagnosing transformers that are still in use, Hong et al. [98] looked at how load current affects transformer winding vibrations. The study simulates frequency responses under various loads while taking insulation properties into consideration, and it finds that load-induced electromagnetic forces have little impact on windings that have been safely clamped. However, as clamping force and current increase, the natural frequency of the windings increases. These findings have been confirmed by tests in the laboratory. They introduced a clamping force indicator, the ratio of natural frequency change to current squared. In particular, normal and degraded transformers can be distinguished from one another by changes in harmonic amplitude with load current. They applied GRU to identify the relationship between current and vibration sequences.

Zhao et al. [99] introduced an improved Generative Adversarial Network (GAN) to improve defect diagnostic performance, particularly for imbalanced data. Compared with normal GANs, this enhanced GAN contains an auxiliary classifier that helps train and use an Auto-Encoder-based technique to estimate the similarity of generated samples. They use an online sample filter to verify that the selected samples meet both accuracy and variation requirements. Their experiments have been carried out with the help of benchmark data from Case Western Reserve University and the XJTU datasets. For dealing with imbalanced fault diagnostic problems, the proposed approach combines GAN, Autoencoder (AE), and Convolutional Neural Network (CNN). CNN converts time-frequency information of original signals into 2D images using Wavelet Transform (WT) based signal processing. To create samples, the one-dimensional data are separated into segments with a segment length of 256 and a sliding step of 64 to match the computing needs and information capacity. They introduced four types of faults in the inner raceway, three outer raceway faults, four rolling element faults and one healthy condition.

## 5. Conclusions and Future Work

### 5.1. Conclusions

The in-depth study of recent research publications on Google Scholar shows the importance of vibration-based condition monitoring as an important tool for ensuring machinery reliability and efficiency across a wide range of industries. The primary goal of our review was to explore the various approaches for processing vibration data within the frame of vibration-based condition monitoring. After reviewing a number of research articles, we were able to determine that vibration data processing techniques can be broadly divided into three primary categories: preprocessing techniques, artificial intelligence algorithms, and signal processing techniques.

The literature review we performed examined signal processing-based procedures that covered domains including time domain, frequency domain, and time-frequency domain studies. In particular, these techniques illustrate the importance that domain-specific knowledge provides successful vibration-based condition monitoring by using techniques like condition curve analysis parameter thresholding, or any other technique to extract useful details from vibration data. While using signal-based processing methods, vibration data are processed and analysed using traditional domain knowledge. Although it is useful in many situations, this approach often relies on human experience and manual intervention, making it unsuitable for completely automated condition monitoring systems.

Preprocessing techniques are important for improving raw vibration data by applying specific filters and then extracting relevant information, such as frequency spectrum or statistical features. This method can be easily integrated with algorithms based on artificial intelligence or with signal processing-based methods, allowing for an accurate method of vibration-based condition monitoring. Artificial intelligence methods, like deep learning and machine learning, provide flexible methods for vibration-based condition monitoring. Machine learning algorithms require feature extraction and selection before using classifiers such as support vector machines, decision trees, or random forests. However, feature selection can be a time-consuming and challenging task. However, deep learning models like convolutional neural networks and recurrent neural networks reduce the need for manual feature engineering, allowing for faster analysis of data for condition monitoring. A limitation of artificial intelligence-based approaches for vibration-based condition monitoring is the imbalanced data, as faulty cases are generally uncommon in real-world data. To address this, anomaly detection algorithms such as one-class support vector machines which are efficient at detecting deviations from normal patterns, play an important role.

### 5.2. Recommendation for Future

In the future, it is recommended to acquire domain knowledge in signal processing and focus should be given to improving data quality prior to treatment in both time and in frequency domains. The most important aspect of an effective vibration-based condition monitoring system is domain-specific information. It provides analysts with an in-depth understanding of the typical behaviour of the machinery as well as distinct vibration signatures connected to different kinds of issues or operating circumstances. Their expertise in analysing vibration data allows them to make distinctions between typical fluctuations and anomalous indications of possible malfunctions. Additionally, domain knowledge also allows analysts to consider contextual aspects such as climate conditions, previous maintenance records and equipment design requirements. This knowledge also helps to optimize monitoring measures, such as sampling frequency, sensor placement as well as analysis approaches based on individual machinery's characteristics. Maintenance workers can use domain expertise to make educated judgements for planning maintenance activities, avoiding catastrophic failures and reducing downtime. This involves extracting statistical features such as mean, median, mode, etc., from processed signals. Using thresholding logic, specific threshold values will be created for these parameters in order to classify vibration signals as healthy or defective. The domain-specific knowledge of signal processing as well as the machinery will help the domain experts to set thresholds accurately. Future studies may examine the time-frequency domain, in which time-domain signals are converted to scalograms or spectrograms, which allows domain specialists to look at them similarly to medical professionals evaluating X-rays to identify abnormalities. The goal is to integrate these approaches into a comprehensive framework for condition monitoring and to validate how effective they are in a variety of industrial applications.

In the future, machine learning methods for vibration-based condition monitoring can be integrated with signal processing and preprocessing methods. One direction for future research might be to study the process of converting signals into the time-frequency domain, like scalograms or spectrograms, and then extract features to compare with those obtained from signals in the time and frequency domains independently. After feature extraction, these extracted features can be used to evaluate the performance of machine learning classifiers. The development of lightweight machine learning or deep learning models can be evaluated to classify vibration signals into classes that correspond to healthy or defective, in the case when there are data with labels. However, one-class support vector machines (OCSVM) will be considered for evaluation in situations where the only available data are healthy and there are no defective instances. Using pre-trained deep learning networks to extract features from healthy data is another direction for the future. The use of anomaly detection methods for fault identification, like One-Class Support Vector Machine (OCSVM), might then be investigated on these learned features. As an anomaly detection

algorithm, the OCSVM is good at detecting deviations from standard, healthy data patterns, which helps to predict failures in vibration-based condition monitoring systems. Anomaly detection algorithms use specialised techniques designed for the identification of unusual events or abnormalities within normal (healthy data sets). They play an important role in helping condition monitoring by overcoming the problem of imbalanced data. Mostly it is very difficult to obtain unhealthy data in datasets from heavy machinery. The dataset collected from heavy machinery for condition monitoring contains healthy data usually. The anomaly detection algorithm (like a one-class support vector machine) tries to detect whether the incoming data show any deviations from the data based on which it is already trained (healthy data). If it shows any deviations from the data that it has seen, then they are considered unhealthy data. Creating synthetic data using data augmentation techniques like the Synthetic Minority Over-sampling Technique (SMOTE) can balance the class distribution.

The availability of unhealthy data is a difficulty in artificial intelligence-based predictive maintenance for heavy machinery, making it a challenging research topic. In the future, the exploration or study using generative artificial intelligence approaches such as Generative Adversarial Networks (GAN) or Variational Autoencoders (VAE) may be carried out. It would be useful to investigate the usage of GANs and VAEs to produce synthetic faulty data from healthy data, as these approaches are capable of understanding the underlying data distribution. By modifying the underlying distribution using predefined rules, it is possible to generate defective data, which can then be used to train machine learning or deep learning models for fault identification.

**Author Contributions:** Conceptualisation, I.U.H.; methodology, I.U.H.; formal analysis, I.U.H.; investigation, I.U.H.; writing—original draft preparation, I.U.H.; writing—review and editing, I.U.H. and K.P.; supervision, K.P.; project administration, K.P. and J.W.; funding acquisition, J.W. All authors have read and agreed to the published version of the manuscript.

**Funding:** This work was supported, in part, by Science Foundation Ireland grant 13/RC/2094_P2 and co-funded under the European Regional Development Fund through the Southern & Eastern Regional Operational Programme to Lero- The Science Foundation Ireland Research Centre for Software www.lero.ie, accessed on 22 March 2024.

**Institutional Review Board Statement:** Not applicable.

**Informed Consent Statement:** Not applicable.

**Data Availability Statement:** No new data were created or analyzed in this study. Data sharing is not applicable to this article.

**Acknowledgments:** The paper and its research would not be made possible without the help of the IMaR team at Munster Technological University.

**Conflicts of Interest:** The authors declare no conflicts of interest.

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
