# Peer review of "Review of Data Processing Methods Used in Predictive Maintenance for Next Generation Heavy Machinery"

_data, 2024_

Round 1

Reviewer 1 Report

Comments and Suggestions for Authors

1.In Section 2.1, the literature on data processing methods based on signal processing is surveyed and how these articles perform signal processing is listed. However, I recommend a simple classification of these articles at the end of each section. And explain the basis of classification instead of making a single list. In addition, in Section 2.1.3, only 5 articles were selected for analysis. Is this analysis comprehensive and in-depth?

2.In Section 4, which is the summary part of the article, the author is still analyzing the research of other people's papers. There is a lack of critical analysis of the mentioned papers. It is difficult to reflect the author's understanding and insight into the research field and to provide readers with new perspectives and thinking. 

3.The innovation of this review article needs to be improved. It is recommended to expand future suggestions based on the survey results. Some new perspectives, new research directions, and new research methods can be proposed in this part. After all, this part is the highlight of the article and can give some research suggestions to subsequent researchers. 

4.The newly published related works such as 'DOI:10.1016/j.ress.2021.107560'  and 'DOI:10.1016/j.ress.2022.108793' should be added and discussed,especially from this journal(Data).

Comments on the Quality of English Language

The paper should be proofread carefully.

Author Response

  1. We have classified in these categories papers based on the techniques used in the reviewed papers. For example, if they used a support vector machine in one research paper, we classified this research paper in the category of supervised machine learning. However, if they use CNN, LSTM, or any other deep learning algorithm, then it is classified as deep learning. , if they use some technique that is based on signal processing in the time domain, then we have classified them in the category of time domain-based signal processing methods. Wherever it is necessary, it is mentioned in red. Wherever it is not mentioned, then the category is first introduced, after which methods are reviewed with some explanation.  In section 2.1.3, only five articles are reviewed. This is because, after gathering papers, when we search keywords on Google Scholar, we get that amount in this category in the reviewed papers.

  2. In Section 4 of the review study, we attempted to present a summary of the conclusions and methodology used in the reviewed papers to provide readers with a better understanding of the larger research landscape. While the focus was on summarizing others' research, we attempted to explain each study within its own context and underline the importance of the approaches used.

  3. We have this updated (the future direction section is improved), and it is highlighted in red in the paper.

  4. Section 2.2 now mentions it, marked with red colour. These are not too relevant to add in the summary section. I’ve mentioned them in the relevant section.

Reviewer 2 Report

Comments and Suggestions for Authors

The manuscript aims to provide information on current developments and future directions in vibration-based condition monitoring. This paper includes the following:

The introduction emphasizes the critical role of vibration-based condition monitoring for heavy machinery reliability.

It highlights the significance of prompt fault identification and preventive maintenance to minimize breakdowns.

The review categorizes vibration data processing approaches into three primary categories for effective analysis.

It underscores the importance of precise and automated fault detection systems to enhance reliability.

The review aims to provide insights into current developments and future directions in vibration-based condition monitoring.

It draws on recent research publications to underscore the widespread importance of vibration-based condition monitoring across industries.

For the paper to fully satisfy the set topic, the authors should include answers to the following questions in the manuscript:

How do heavy machinery investments relate to the importance of vibration-based condition monitoring?

What are the primary categories into which vibration data processing approaches are divided in this review?

What role does domain-specific knowledge play in successful vibration-based condition monitoring?

How do preprocessing techniques contribute to improving raw vibration data?

What advantages do artificial intelligence methods offer for vibration-based condition monitoring?

How do anomaly detection algorithms address the challenge of imbalanced data in condition monitoring?

Author Response

Thank you for your feedback. please see answers to your questions below.

How do heavy machinery investments relate to the importance of vibration-based condition monitoring?

It is now added (which is section 3.1) and is marked with red colour.

What are the primary categories into which vibration data processing approaches are divided in this review?

Broadly speaking, it is first categorized into either Signal Processing-based Based Methods (2.1), Artificial intelligence-based Based Methods (2.2). Then, it is also categorized in subcategories. In Signal processing Based Methods (2.1), it is classified into either Time Domain Based Methods (2.1.1), Frequency Domain Based Methods (2.1.2), Time and Frequency Domain Based Methods (2.1.3), Preprocessing Methods for Improving Signals Quality for Fault Detection (2.1.4). While in Artificial intelligence Based Methods (2.2), it is classified into Machine Learning Based Methods (2.2.1) and Deep Learning Based Methods (2.2.2).

What role does domain-specific knowledge play in successful vibration-based condition monitoring?

This is now discussed in the first paragraph of future work (section 5.2).

How do preprocessing techniques contribute to improving raw vibration data?

It is now added, which is section 3.5, and is marked with red Color.

What advantages do artificial intelligence methods offer for vibration-based condition monitoring?

It is already mentioned in section 3.6, highlighted in red colour.

How do anomaly detection algorithms address the challenge of imbalanced data in condition monitoring?

This is now added in the second paragraph of future work (section 5.2).